# Towards Distribution-Agnostic Generalized Category Discovery

**Jianhong Bai**[1], **Zuozhu Liu**[1✉], **Hualiang Wang**[2], **Ruizhe Chen**[1], **Lianrui Mu**[1],
**Xiaomeng Li**[2], **Joey Tianyi Zhou**[3], **Yang Feng**[4], **Jian Wu**[1], **Haoji Hu**[1✉]
[1]Zhejiang University, [2]The Hong Kong University of Science and Technology,
[3]Centre for Frontier AI Research, A*STAR, [4]Angelalign Technology Inc.
jianhongbai@zju.edu.cn

## Abstract

Data imbalance and open-ended distribution are two intrinsic characteristics of the real visual world. Though encouraging progress has been made in tackling each challenge separately, few works dedicated to combining them towards real-world scenarios. While several previous works have focused on classifying close-set samples and detecting open-set samples during testing, it's still essential to be able to classify unknown subjects as human beings. In this paper, we formally define a more realistic task as distribution-agnostic generalized category discovery (DA-GCD): generating fine-grained predictions for both close- and open-set classes in a long-tailed open-world setting. To tackle the challenging problem, we propose a Self-**Ba**lanced **Co**-Advice co**n**trastive framework (BaCon), which consists of a contrastive-learning branch and a pseudo-labeling branch, working collaboratively to provide interactive supervision to resolve the DA-GCD task. In particular, the contrastive-learning branch provides reliable distribution estimation to regularize the predictions of the pseudo-labeling branch, which in turn guides contrastive learning through self-balanced knowledge transfer and a proposed novel contrastive loss. We compare BaCon with state-of-the-art methods from two closely related fields: imbalanced semi-supervised learning and generalized category discovery. The effectiveness of BaCon is demonstrated with superior performance over all baselines and comprehensive analysis across various datasets. Our code is available at: https://github.com/JianhongBai/BaCon.

## 1 Introduction

Data imbalance [82, 79, 45] and open-ended distribution [14, 72, 17] are inherently intertwined with each other in the real visual world [40]. On the one hand, the frequency of different objects exhibits long-tailed distribution [48, 54] where a small portion of classes dominate the distribution and many classes are associated with only a few examples, as observed in healthcare [42, 75], autonomous driving [41, 36] and species classification [59]. On the other hand, humans or intelligent agents constantly encounter novel visual instances. Such complicated intertwined context poses great challenges to develop effective visual algorithms [28, 55] that apply to real-world scenarios [1, 12].

Tremendous efforts have been devoted to addressing the challenges of long-tail and open-set learning. In long-tail learning (LTL), previous works aim to highlight minority categories by re-sampling [53, 83, 16] or re-weighting [5, 26] approaches, debiasing model prediction from a statistical perspective [49, 43], decoupling feature learning and classifier training [29], or generalizing to semi-supervised [33, 31, 69] and self-supervised [38, 27, 81] scenarios. For open-world learning, existing work can be divided into three sub-domains based on how to handle open-set samples. Robust semi-supervised

---

✉ Corresponding authors.

37th Conference on Neural Information Processing Systems (NeurIPS 2023).

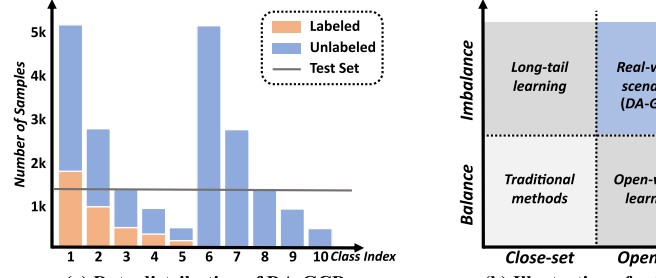
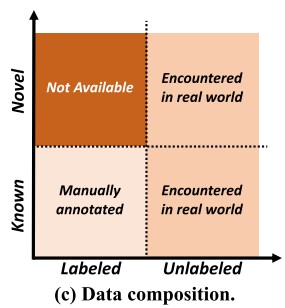

(a) Data distribution of DA-GCD.     (b) Illustration of settings.     (c) Data composition.

Figure 1: Illustration of distribution-agnostic generalized category discovery (DA-GCD).

learning (SSL) [46] assumes that out-of-distribution (OOD) samples may undesirably appear in the unlabeled set, and several techniques [20, 77, 9] are designed to *reject* these potential open-set samples to improve model robustness. Another line of methods [3, 23, 37, 34, 61, 44, 64] attempt to *detect* them with close-set knowledge during testing and formulate the task as OOD detection. Furthermore, generalized category discovery (GCD) [60] suggests to *classify* open-set samples into each class [4, 56, 70]. An earlier work OLTR [40] manage to integrate LTL and open-world learning towards more realistic visual recognition. OLTR handles close-set data imbalance and is aware of open-set instances similar to OOD detection. However, this approach does not meet the requirements for real-world recognition scenarios as: a) it works in a fully-supervised manner and does not effectively utilize a large amount of unlabeled data in the real world. b) it can only *detect* unseen (open-set) samples rather than classifying them with novel visual concepts.

In this work, we consider a real-world yet more challenging setting: we assume that only a portion of classes are collected and manually annotated (known classes) while massive unlabeled data from both known and open-set categories (novel classes) are available in the real world (Figure 1c). These two types of data, both long-tailed, form the semi-supervised training set, and the model is expected to *classify* **all** known and novel classes in a balanced test set, as illustrated in Figure 1. We name this setting Distribution-Agnostic Generalized Category Discovery (DA-GCD) and contrast it with other related settings in Table 1. DA-GCD differs from previous open-world tasks because it considers realistic long-tail settings. It also differs from the pioneering work OLTR [40] by further considering the semi-supervised setting and classifying novel classes, as in up-to-date open-world settings [60, 4].

The DA-GCD setting presents significant challenges that cannot be resolved with existing methods. Firstly, as confirmed by numerous studies in the field of long-tail learning, imbalanced datasets lead to severe performance degradation and model bias [79, 29, 27]. This issue is even more challenging in DA-GCD, where the prior dataset distribution is unavailable in open-world scenarios. Therefore, most LTL methods [43, 50] could not be directly extended to resolve the DA-GCD task. Meanwhile, since only unlabeled samples are available for novel classes (as shown in Figure 1c), the performance on these open-set classes is severely limited. Existing contrastive-based methods in GCD [60, 56] struggle to optimize the feature space of unlabeled samples, resulting in poor alignment and uniformity [66]. Lastly, GCD methods lack tailored designs for imbalanced datasets, leading to significant performance degradation on long-tailed datasets (as discussed in Section 5.2).

To conquer the aforementioned challenges, we propose BaCon, a novel Self-**Ba**lanced **Co**-Advice co**n**trastive framework. BaCon comprises a contrastive-learning branch and a pseudo-labeling branch, which work collaboratively to provide interactive supervision to tackle the imbalanced recognition task under open-world scenarios. Specifically, the contrastive-learning branch provides distribution estimation during training, which is used to regularize the prediction results of the linear classifier for better pseudo-labeling. On the other hand, the generated pseudo-labels are further sampled and debiased to re-balance and provide additional supervision for the contrastive-learning branch. To fuse the knowledge of the two branches and learn a better feature space, we design a novel pseudo-label-based contrastive loss that clusters samples based on their *positiveness* scores.

We make the following major contributions. **1)** We formulate a more realistic DA-GCD task that unifies long-tail and open-world learning and identify unique challenges that cannot be adequately addressed by existing techniques in both fields. **2)** We design a novel self-**Ba**lanced **Co**-advice co**n**trastive (BaCon) framework for DA-GCD, which comprises a contrastive-learning branch and pseudo-labeling branch that work collaboratively to tackle the data imbalance and classify novel (open-set) categories simultaneously. **3)** We conduct extensive experiments on various datasets and perform fine-grained comparisons with SOTA baselines to validate the effectiveness of BaCon.

Table 1: Comparison of DA-GCD with other similar settings.

| Setting | Scenario | Known classes | Novel classes | Involved in training? (novel classes) | Data Distribution |
|---|---|---|---|---|---|
| Robust SSL | Semi-supervised | Classify | Reject | ✓ | Balanced |
| OOD Det | Depend on method | Classify | Detect | Depend on method | Balanced |
| GCD | Semi-supervised | Classify | Classify | ✓ | Balanced |
| LT-SSL | Semi-supervised | Classify | N.A. | ✗ | Long-tailed |
| OLTR | Fully-supervised | Classify | Detect | ✗ | Long-tailed |
| **DA-GCD** | Semi-supervised | Classify | Classify | ✓ | Long-tailed |

## 2  Related Works

**Open-World Learning** tackling the scenario where some unknown (open-set) visual concepts or categories are involved during training or real-world applications. OOD detection [37, 24, 39], as a well-known sub-field, considering a distributional shift (either semantic shift [23] or covariate shift [35]) between the train and test stage, with the goal of *detecting* these OOD samples during testing. Robust semi-supervised learning [46, 20, 77, 9] apply different techniques to *reject* undesirable OOD samples in the unlabeled set to improve model robustness. More recently, a line of work [60, 4, 56, 70, 52, 51] propose to *classify* the open-set samples according to their visual concepts. Nevertheless, these methods are built on the class balance assumption, and we observe a consistent performance degradation when generalizing to DA-GCD scenarios. Distribution-agnostic recognition is also investigated by [73], which focuses on the task of novel category discovery.

**Long-Tail Learning** refers to a scenario where the majority of samples belong to a few classes, while the minority classes have only a small number of samples [63, 15]. Early research on long-tail learning mainly focused on fully-supervised scenarios, with representative methods that highlight the minority samples via re-sampling [53, 83, 16], re-weighting [5, 26], or knowledge transferring [68, 11, 76]. Long-tail learning under semi-supervised [33, 31, 69] and self-supervised [2, 74, 38, 27, 81, 78] settings are also widely investigated and have notable achievements. However, the requirement of the dataset distribution obstructs the application of fully/semi-supervised methods in DA-GCD, while self-supervised methods would disregard the available label information for known samples.

**Contrastive Learning** [8, 22, 18] provide distinctive and transferable representations by controlling the instance similarity in feature space, achieving significant progress in recent years. The following literature generalizes the idea to different scenarios [30, 62, 27], implements it for solving various downstream tasks [7, 71], and provides theoretical support [66, 67]. However, few attempts have been made at contrastive-based open-world recognition. GCD [60] proposes performing supervised CL on labeled samples and unsupervised CL on all data. OpenCon [56] generating pseudo-positive pairs for closely aligned representations, but this paradigm could lead to another dilemma: the representations and pseudo-labels are interdependent, which means an inferior feature space (e.g., lack of intra-class consistency) could lead to false positive pairs, and in turn deteriorate the learning of feature space. In short, those methods are incompatible with the proposed DA-GCD setting and have inferior representations for samples belonging to *novel* classes.

## 3  Preliminaries

### 3.1  Problem Formulation

As illustrated in Figure 1a, our training set is composed of a labeled subset $\mathcal{D}^l = \left\{ (\boldsymbol{x}_i^l, y_i^l) \right\} \in \mathcal{X} \times \mathcal{Y}_l$ and an unlabeled subset $\mathcal{D}^u = \left\{ (\boldsymbol{x}_i^u, y_i^u) \right\} \in \mathcal{X} \times \mathcal{Y}_u$, where we don't have the label information for **any** images in novel classes, i.e., $\mathcal{Y}_l = \{\mathcal{Y}_k\}$, $\mathcal{Y}_u = \mathcal{Y} = \{\mathcal{Y}_k \cup \mathcal{Y}_n\}$. Both $\mathcal{D}^l$ and $\mathcal{D}^u$ present a long-tail distribution with imbalance ratio $\rho_l, \rho_u \gg 1$. The objective of DA-GCD is to assign labels on a disjoint balanced test set containing all classes in $\mathcal{Y}_u$. The total number of classes in the whole dataset $\mathcal{D}$ is commonly regarded as a known prior [60, 4, 70], as we can effectively estimate it through off-the-shelf methods [21].

## 3.2 Revisiting Contrastive Learning

Contrastive-based representation learning methods [8, 22, 18, 30] aim to find an embedding function $f$ to acquire optimal feature representation $\boldsymbol{z} \in \mathbb{R}^d$ of the input image $\boldsymbol{x} \in \mathbb{R}^{CHW}$ with $\boldsymbol{z} = f(\boldsymbol{x})$, such that $\boldsymbol{z}$ retains the discriminative semantic information of the input image. The general contrastive learning loss can be defined as:

$$\mathcal{L}_{\text{CL}}(\mathcal{D}) = \frac{1}{|\mathcal{D}|} \sum_{i \in \mathcal{D}} \left[ \frac{1}{|P(i)|} \sum_{p \in P(i)} -\log \frac{\exp(\boldsymbol{z}_i \cdot \boldsymbol{z}_p / \tau)}{\sum_{a \in A(i)} \exp(\boldsymbol{z}_i \cdot \boldsymbol{z}_a / \tau)} \right], \tag{1}$$

where $\mathcal{D}$ is the training set. $P(i)$ is the set of indices of positive pairs in $A(i)$, $|\cdot|$ denotes the operation to compute the cardinality, and $\tau$ is a hyper-parameter. In unsupervised learning, the positive set is formulated as the two views of the same image [8, 22]. [30] further generalizes the contrastive loss into supervised scenarios where the positive set is the images that share the same ground truth label.

# 4 Self-Balanced Co-Advice Contrastive Framework

As illustrated in Figure 2, we design a two-branch structure framework for DA-GCD. In Section 4.1, we propose the contrastive-learning branch to estimate the training distribution and further regularize the outputs of the pseudo-labeling branch. We then describe techniques for self-balanced knowledge transfer from the pseudo-labeling branch to the contrastive-learning branch in Section 4.2. Finally, in Section 4.3, a novel contrastive loss is proposed for better novel class learning.

## 4.1 Regularize Predictions with Dynamic Distribution Estimation

To enhance and re-balance the contrastive representation learning, we propose to learn an auxiliary classifier for pseudo-labeling. Our pseudo-labeling branch includes a cross-entropy loss $\mathcal{L}_s$ on labeled data, a semi-supervised objective $\mathcal{L}_u$ (e.g., cross pseudo supervision in [70]) on all samples, and a regularization term $\mathcal{L}_{reg}$ to align the predictions with the data distribution to avoid non-activated classifiers. However, the distribution of the training set is agnostic in DA-GCD, and simply using the distribution of labeled set $\boldsymbol{\pi}_{\mathcal{D}^l}$ or a balance prior results in inferior performance (Table 6a). Furthermore, we observed that estimating the distribution from the pseudo-labeling branch itself could accumulate the estimation error and deteriorate model performance.

To this end, we perform estimation on the contrastive-learning branch as an alternative to avoid the accumulation of estimation bias. Concretely, we perform k-means clustering on all samples in the training set $\mathcal{D}$ to obtain $C$ clusters with size $\boldsymbol{n}$, where $C = |\mathcal{Y}|$, and normalized the sample number $\boldsymbol{n}$ to frequency $\boldsymbol{\pi}_e$, i.e., $\boldsymbol{\pi}_e = \boldsymbol{n} / \sum_{c=1}^{C} (n[c])$. Moreover, we also need to determine the corresponding relationship between clusters and categories. Therefore, we use the Hungarian optimal assignment algorithm [32] to map $|\mathcal{Y}^l|$ clusters to each known class. For the remaining $|\mathcal{Y}| - |\mathcal{Y}^l|$ clusters, we sort them by the cluster size and assign them sequentially to the novel classes.[1] Finally, we regularize the mean prediction of the pseudo-labeling branch with the aligned estimated distribution $\boldsymbol{\pi}_e$:

$$\mathcal{L}_{reg} = \text{KL} \left[ \frac{1}{|B|} \sum_{i \in B} \text{softmax}(f_{cls}(\boldsymbol{x}_i)) \,\|\, (\text{align}(\boldsymbol{\pi}_e))^p \right], \tag{2}$$

where $B$ refers to the batch size during training, $f_{cls}$ denote the backbone and the linear classifier of the auxiliary branch, $\text{KL}[\cdot \| \cdot]$ is the Kullback-Leibler (KL) divergence between the two distributions, and $p \in [0, 1]$ is a hyper-parameter used to smooth the target long-tailed distribution. To balance estimation accuracy and computation overhead, we re-estimate the dataset distribution every $r$ epoch. The overall training objective of the pseudo-labeling branch is as follows:

$$\mathcal{L}_{cls} = \mathcal{L}_s + \eta_1 \mathcal{L}_u + \eta_2 \mathcal{L}_{reg}. \tag{3}$$

---

[1]During inference, we use optimal assignment to determine the correspondence between classifiers and categories (detailed in Section 5.1), hence sequentially assigning the clusters to classes will not affect performance.

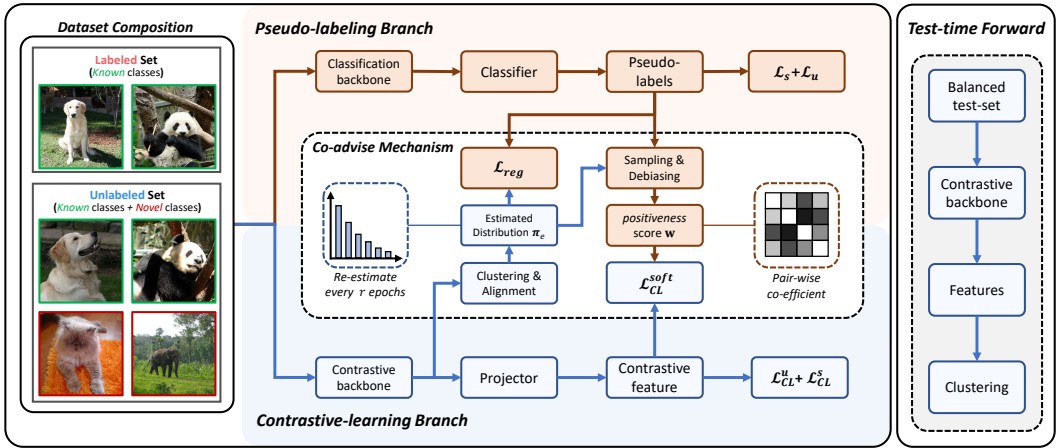

Figure 2: Overview of the self-balanced co-advice contrastive framework (BaCon).

## 4.2 Self-Balanced Knowledge Transfer: Debiasing and Sampling

In Section 4.1, we regularize the pseudo-labeling branch with the estimated dataset distribution $\boldsymbol{\pi}_e$ acquired from the contrastive-learning branch. In this step, we aim to transfer the knowledge from the pseudo-labeling branch to the contrastive-learning branch in turn and further enhance the representation learning, which is also beneficial to the estimation of data distribution $\boldsymbol{\pi}$. However, the long-tail distribution of the underlying dataset places additional requirements on knowledge transferring. On one hand, we should ensure that the knowledge to be transmitted is not affected by the long-tailed distribution. Meanwhile, we hope that this process can help the contrastive-learning branch cope with the issue of data imbalance and the lack of supervision for novel classes.

We design a debiasing and sampling step of the pseudo-labels to meet the two aforementioned requirements respectively for knowledge transfer. First, we apply post-hoc logits adjustment [43] based on the estimated distribution to the predicted logits $f_{cls}(\boldsymbol{x}_i)$ to eliminate the bias caused by long-tailed distribution:

$$\widetilde{\boldsymbol{p}}_i = \mathrm{softmax}(f_{cls}(\boldsymbol{x}_i) - k \cdot \log\boldsymbol{\pi}_e)), \tag{4}$$

where $\widetilde{\boldsymbol{p}}_i$ denotes the rectified class probability prediction of sample $\boldsymbol{x}_i$, and $k$ is a hyper-parameter in [43]. Next, in order to re-balance the learning process and filter low-precision pseudo-labels, we propose to sample the unlabeled instances. Specifically, we sample the pseudo-labels of unlabeled instances in a training batch $\widetilde{\boldsymbol{p}}_B = \{\widetilde{\boldsymbol{p}}_i : i = 1, \cdots, B\}$ according to their prediction class $c$, where $c_i = \arg\max \widetilde{\boldsymbol{p}}_i$. Formally, the class-wise sampling rate $\mathrm{SR}^c$ can be defined as:

$$\mathrm{SR}^c = \begin{cases} (\frac{\boldsymbol{\pi}_e^c}{\min(\boldsymbol{\pi}_e)})^{-\alpha}, & c \in \mathcal{Y}_B, \\[2mm] (\frac{\boldsymbol{\pi}_e^c}{\min(\boldsymbol{\pi}_e)})^{-\beta}, & \text{otherwise}, \end{cases} \tag{5}$$

where $\boldsymbol{\pi}_e$ is the estimated distribution introduced in Section 4.1, $\mathcal{Y}_B \subseteq \mathcal{Y}_k$ denotes the set of labels which involved in the current batch, and $\alpha, \beta \in [0, 1]$ are two hyper-parameters. Setting $\alpha = \beta = 0$ means sampling all pseudo-labels, while $\alpha = \beta = 1$ means setting the sampling rate to be inversely proportional to the estimated number of samples in its category. Note that we prioritize selecting samples with higher prediction confidence in each class to remove the potentially false pseudo-labels. The sampled instances $\{M(\mathcal{D}_u)\}$ complement the original long-tailed distribution to serve as a re-balance term for the labeled subset $\{\mathcal{D}_l\}$ and also provide additional supervision for *novel* classes. With this debiasing and sampling step, the sampled instances $\{\mathcal{D}_l \cup M(\mathcal{D}_u)\}$ and their corresponding rectified pseudo-labels mitigate the impact of the imbalance issue and compensate the unlabeled open-set samples simultaneously.

### 4.3 Pseudo-label-based Soft Contrastive Learning

In Section 4.1 and Section 4.2, we generate pseudo-labels via the pseudo-labeling branch with help from the contrastive-learning branch. Those pseudo-labels could provide additional information to improve the contrastive-learning branch as well. In particular, we adopt a *soft* contrastive strategy to fully leverage the probabilistic information in pseudo-labels. We design a pair-wise *positiveness* score to adjust the contribution of different samples to the anchor instance. For image pair $(\boldsymbol{x}_i, \boldsymbol{x}_j)$, the *positiveness* score $w_{ij} = \text{Sim}(\widetilde{\boldsymbol{p}}_i, \widetilde{\boldsymbol{p}}_j)$. $w_{ij}$ is obtained by calculating the similarity of the rectified class probability distribution between $\boldsymbol{x}_i$ and $\boldsymbol{x}_j$. In practice, We implement the similarity metric with the dot product operation, which can be mathematically interpreted as the probability of instance $\boldsymbol{x}_i$ and $\boldsymbol{x}_j$ belonging to the same class. We compare different definitions of the *positiveness* score in Section 5.4. Finally, we formulate the soft contrastive loss by incorporating $\boldsymbol{w}$ into Eq. 1:

$$\mathcal{L}_{\text{CL}}^{soft}(\mathcal{D}) = \frac{1}{|\mathcal{D}|} \sum_{i \in \mathcal{D}} \left[ \frac{1}{\sum_{j \in A(i)} w_{ij}} \sum_{j \in A(i)} -w_{ij} \cdot \log \frac{\exp(\boldsymbol{z}_i \cdot \boldsymbol{z}_j / \tau)}{\sum_{a \in A(i)} \exp(\boldsymbol{z}_i \cdot \boldsymbol{z}_a / \tau)} \right]. \quad (6)$$

**Claim 1.** *Assume we train a model $\theta$ with the proposed soft contrastive loss, and $\boldsymbol{x}_i$ and $\boldsymbol{x}_j$ are two samples within a training batch B. The optimal value of the contrastive logit $p_{ij}^*$ is $\frac{w_{ij}}{\sum_{k=1}^{|B|} w_{ik}}$, where $w_{ij}$ is the corresponding positiveness score for the sample pair $(\boldsymbol{x}_i, \boldsymbol{x}_j)$ in Eq. 6.*

The proof can be found in Appendix A. Claim 1 indicates that minimizing Eq. 6 encourages the similarity of features between two samples to be proportional to the corresponding *positiveness* score. In this way, we effectively transfer knowledge from the pseudo-labeling branch to contrastive learning. The overall training objective of the contrastive-learning branch consists of an unsupervised CL loss on all samples, a supervised CL loss on labeled instances, and the proposed soft CL loss on both labeled and sampled unlabeled subset:

$$\mathcal{L}_{con} = \mathcal{L}_{\text{CL}}^u(\mathcal{D}) + \gamma_1 \mathcal{L}_{\text{CL}}^s(\mathcal{D}_l) + \gamma_2 \mathcal{L}_{\text{CL}}^{soft}(\mathcal{D}_l \cup M(\mathcal{D}_u)). \quad (7)$$

The pipeline of our method is illustrated in Figure 2, and the pseudo-code of BaCon can be found in Appendix B. It's worth noting that the backbone of the two branches is shared and frozen during training (except for the last block). Therefore the two-branch structure only slightly increases computational overheads. During inference, we only utilize the backbone of the contrastive-learning branch and obtain predictions via k-means clustering on the backbone features.

## 5 Expriments

### 5.1 Experimental Setup

**Data Preparation** We conduct experiments on four popular datasets. CIFAR-10-LT/CIFAR-100-LT are long-tail subsets sampled from the original CIFAR10/CIFAR100 [10]. We set the imbalance ratio to 100 in default. ImageNet-100-LT is proposed by [27] with 12K images sampled from ImageNet-100 [57] with Pareto distribution. Places-LT [40] contains about 62.5K images sampled from the large-scale scene-centric Places dataset [80] with Pareto distribution. For each dataset, we first sub-sample $|\mathcal{Y}_k|$ classes from the whole long-tail datasets to stimulate known classes and treat the rest as novel classes, then sub-sample 50% instances in each known class as $\mathcal{D}^l$ and concentrate others with all samples in novel classes to form $\mathcal{D}^u$. The default $|\mathcal{Y}_k|$ for CIFAR-10-LT, CIFAR-100-LT, ImageNet-100-LT, Places-LT are 5, 80, 50, 182 respectively. We also evaluate BaCon with different ratios of $|\mathcal{Y}_k|$ and $|\mathcal{Y}_n|$, set up $\mathcal{D}^l$ with few annotated samples, $\rho_l \neq \rho_u$ in E.

**Evaluation Protocols and Metrics** Following previous works [60, 4], we calculate the accuracy by assigning the model prediction (cluster result) $\hat{\boldsymbol{y}}$ to ground truth $\boldsymbol{y}$ via the Hungarian assignment algorithm [32]. With the assignment fixed, we report the overall classification accuracy on $\mathcal{Y}_k$ (Old), $\mathcal{Y}_n$ (New), and $\mathcal{Y}_k \cup \mathcal{Y}_n$ (All). To observe the impact of long tail distribution on performance, we evenly divide $\mathcal{Y}_k$ and $\mathcal{Y}_n$ into three disjoint groups {*Many, Median, Few*} respectively in terms of the number of instances of each class. We also calculate the standard deviation (Std) of the accuracy of the three groups, which quantitatively analyze the balancedness of a feature space [27].

Table 2: Test accuracy (%) and balancedness (Std↓) of existing methods on CIFAR-100-LT.

| Method | Subset | Many↑ | Med↑ | Few↑ | Std↓ | All↑ | Many↑ | Med↑ | Few↑ | Std↓ | All↑ | Overall |
|---|---|---|---|---|---|---|---|---|---|---|---|---|
| | | Known | | | | | Novel | | | | | Overall |
| GCD | $\mathcal{D}_b$ | 66.2 | 75.4 | 73.6 | 4.0 | 71.8 | 60.3 | 54.1 | 51.5 | 3.7 | 55.2 | 68.5 |
| | $\mathcal{D}_i$ | 75.9 | 69.4 | 52.9 | 9.7 | 65.5 | 41.9 | 54.2 | 49.8 | 5.1 | 49.0 | 62.2 |
| SimGCD | $\mathcal{D}_b$ | 75.1 | 75.3 | 72.2 | 1.4 | 74.2 | 60.2 | 58.8 | 59.7 | 0.6 | 59.5 | 71.3 |
| | $\mathcal{D}_i$ | 72.8 | 71.1 | 34.6 | 17.6 | 59.8 | 32.9 | 25.8 | 15.5 | 7.1 | 24.8 | 52.8 |
| ABC | $\mathcal{D}_i$ | 77.4 | 51.6 | 16.3 | 25.0 | 48.5 | 20.8 | 32.0 | 5.9 | 10.7 | 20.8 | 43.0 |
| DARP | $\mathcal{D}_i$ | 74.8 | 55.2 | 33.4 | 16.9 | 54.5 | 29.9 | 30.1 | 10.0 | 9.4 | 24.0 | 48.4 |

Table 3: Test accuracy (%) on four generic long-tailed image recognition datasets. (**bold**: best performance among all methods, underline: best performance among the baseline methods.)

| Methods | CIFAR-10-LT | | | CIFAR-100-LT | | | ImageNet-100-LT | | | Places-LT | | |
|---|---|---|---|---|---|---|---|---|---|---|---|---|
| | Old | New | All | Old | New | All | Old | New | All | Old | New | All |
| ABC[†] | 77.7 | 20.2 | 48.9 | 48.5 | 20.8 | 43.0 | - | - | - | - | - | - |
| DARP[†] | 77.6 | 35.2 | 56.4 | 54.5 | 24.0 | 48.4 | - | - | - | - | - | - |
| TRSSL[†] | 78.4 | 66.8 | 72.6 | 58.7 | 35.8 | 54.1 | - | - | - | - | - | - |
| ORCA[†] | 77.5 | 55.6 | 66.6 | 55.0 | 30.8 | 50.1 | 69.3 | 29.0 | 49.2 | 21.5 | 6.9 | 14.2 |
| SimGCD | 75.1 | 41.5 | 58.3 | 59.8 | 24.6 | 52.8 | 81.1 | 33.4 | 57.8 | **31.4** | 18.2 | 24.8 |
| GCD | 78.5 | 71.7 | 75.1 | 65.5 | 49.0 | 62.2 | 81.0 | 76.8 | 78.9 | 29.8 | 22.7 | 26.2 |
| OpenCon[†] | 87.2 | 47.2 | 67.2 | 64.2 | 40.9 | 59.6 | 83.3 | 42.8 | 63.0 | 30.6 | 12.4 | 21.6 |
| BaCon-O | 83.3 | 78.0 | 80.7 | 66.5 | **69.6** | 67.1 | 84.2 | 80.0 | 82.1 | 30.7 | 25.6 | 28.1 |
| BaCon-S | **94.2** | **88.1** | **91.1** | **67.4** | 66.5 | **67.2** | **84.6** | **82.8** | **83.7** | 31.1 | **28.4** | **29.9** |

**Training Settings** Following GCD [60], we load a DINO [6] pre-trained ViT-B/16 [13] as the backbone for both branches and only fine-tune the last block for all datasets. The classification head and projector are randomly initialized. To achieve a fair comparison, we replace the backbone model of all baseline methods with the ViT pre-trained by DINO (the same as ours), and † denotes adapted methods. More training details can be found in Appendix C.

## 5.2 Migration of Existing Methods to DA-GCD

Here we first state that migrating existing methods from two similar scenarios (GCD [60] and imbalanced SSL [31]) to the proposed DA-GCD result in inferior performance due to the co-occurrence of data imbalance and open-set samples. Existing GCD methods, whether based on classifier [60] or non-parametric classification methods [70], are insufficient in addressing the issue of data imbalance. In Table 2, we report two SOTA methods (GCD [60] and SimGCD [70]) performances on the balanced set $\mathcal{D}_b$ and its long-tailed counterpart $\mathcal{D}_i$. It's observed that they exhibit significant performance degradation (especially on minority classes) on $\mathcal{D}_i$ compared to their performance on $\mathcal{D}_b$. While imbalanced SSL methods [33, 31] can barely classify novel class samples under the DA-GCD setting.

## 5.3 BaCon's Accuracy, Balancedness and Versatility

The results of the BaCon in various datasets are presented in Table 3. BaCon-O and BaCon-S refer to training the pseudo-labeling branch in ORCA-style [4] and SimGCD-style [70] respectively. We compare BaCon with two SOTA imbalanced-SSL methods ABC [33] and DARP [31], and five latest open-world recognition techniques including contrastive-based methods GCD [60] and OpenCon [56] and classifier-based methods TRSSL [52], ORCA [4] and SimGCD [70]. [2] As shown in Table 3, BaCon significantly outperforms all these methods by a large margin under various datasets. Specifically, on CIFAR-100-LT, BaCon-S outperforms the best baseline by 17.5% and 5.0% on *novel* classes accuracy and *overall* accuracy respectively.

---

[2]We don't compare BaCon with OLTR [40] since they perform in supervised scenarios without the presence of novel classes (Table 1), resulting in inferior performance on novel class classification.

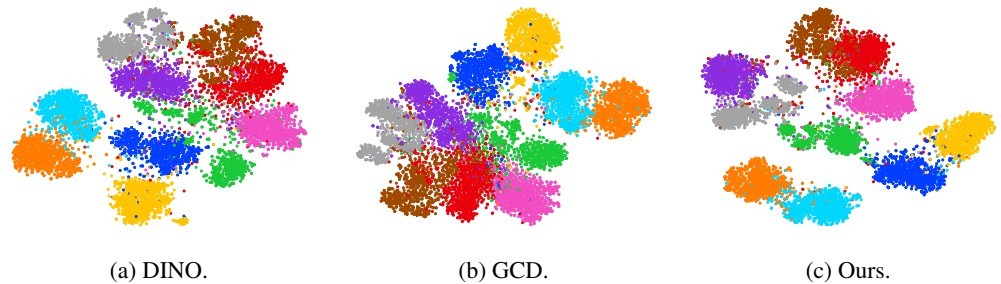

(a) DINO.        (b) GCD.       (c) Ours.

Figure 3: t-SNE visualization on the test set of CIFAR-10.

Table 4: Test accuracy (%) and balancedness (Std↓) on CIFAR-100-LT.

| Methods | Known | | | | | Novel | | | | | Overall |
| | Many↑ | Med↑ | Few↑ | Std↓ | All↑ | Many↑ | Med↑ | Few↑ | Std↓ | All↑ | |
|---|---|---|---|---|---|---|---|---|---|---|---|
| ABC[†] | 77.4 | 51.6 | 16.3 | 25.0 | 48.5 | 20.8 | 32.0 | 5.9 | 10.7 | 20.8 | 43.0 |
| DARP[†] | 74.8 | 55.2 | 33.4 | 16.9 | 54.5 | 29.9 | 30.1 | 10.0 | 9.4 | 24.0 | 48.4 |
| TRSSL[†] | 78.8 | **73.0** | 24.1 | 24.3 | 58.7 | 32.7 | 50.9 | 18.8 | 13.1 | 35.8 | 54.1 |
| ORCA[†] | 73.6 | 60.0 | 31.0 | 17.8 | 55.0 | 47.5 | 28.4 | 17.2 | 12.5 | 30.8 | 50.1 |
| SimGCD | 71.1 | 72.8 | 34.6 | 17.6 | 59.8 | 32.9 | 25.8 | 15.5 | 7.1 | 24.6 | 52.8 |
| GCD | 75.9 | 69.4 | _52.9_ | _9.7_ | _65.5_ | 41.9 | _54.2_ | _49.8_ | _5.1_ | _49.0_ | _62.2_ |
| OpenCon[†] | _86.8_ | 69.7 | 35.8 | 21.2 | 64.2 | 55.8 | 48.9 | 15.3 | 17.7 | 40.9 | 59.6 |
| BaCon-O | 72.9 | 64.7 | **62.0** | **4.6** | 66.5 | 72.6 | **70.6** | **65.3** | **3.1** | 69.6 | 67.1 |
| BaCon-S | 74.4 | 67.1 | 60.8 | 5.6 | **67.4** | **73.7** | 66.1 | 59.8 | 5.7 | **66.5** | **67.2** |

Table 5: Test accuracy (%) on CIFAR-100-LT with different imbalance ratio $\rho$.

| Methods | CIFAR-100-LT ($\rho_l = \rho_u = \rho$) | | | | | | | | | | | |
| | $\rho = 20$ | | | $\rho = 50$ | | | $\rho = 100$ | | | $\rho = 150$ | | |
| | Old | New | All | Old | New | All | Old | New | All | Old | New | All |
|---|---|---|---|---|---|---|---|---|---|---|---|---|
| ABC[†] | 64.0 | 23.0 | 55.8 | 53.7 | 21.5 | 47.3 | 48.5 | 20.8 | 43.0 | 41.9 | 18.5 | 37.2 |
| DARP[†] | 66.8 | 27.4 | 58.9 | 58.9 | 25.1 | 52.1 | 54.5 | 24.0 | 48.4 | 49.7 | 23.3 | 44.3 |
| TRSSL[†] | 71.8 | 43.6 | 66.2 | 62.9 | 49.0 | 60.1 | 58.7 | 35.8 | 54.1 | 55.2 | 33.3 | 50.8 |
| ORCA[†] | 66.9 | 37.7 | 61.0 | 61.7 | 30.0 | 55.3 | 55.0 | 30.8 | 50.1 | 51.0 | 36.5 | 48.1 |
| SimGCD | _71.8_ | 23.1 | 62.1 | 63.7 | 23.4 | 55.7 | 59.8 | 24.2 | 52.8 | 57.5 | 19.9 | 50.0 |
| GCD | 69.8 | _58.7_ | 67.6 | 66.6 | _59.7_ | 65.2 | _65.5_ | _49.0_ | _62.2_ | 65.4 | _50.2_ | _62.4_ |
| OpenCon[†] | 76.3 | 44.3 | _69.9_ | _71.2_ | 42.0 | _65.4_ | 64.2 | 40.9 | 59.6 | 59.7 | 45.3 | 56.8 |
| BaCon-O | 71.5 | 67.9 | 70.8 | 70.0 | 63.7 | 68.7 | 66.5 | **69.6** | 67.1 | 67.4 | 64.2 | 66.8 |
| BaCon-S | 71.6 | **68.5** | **71.0** | 70.4 | **64.2** | **69.2** | 67.4 | 66.5 | **67.2** | 67.0 | 64.3 | 66.5 |

To better evaluate the impact of dataset imbalance, we make a fine-grained comparison in Table 4. The lower standard deviation of the performance of {*Many*, *Median*, *Few*} on both known and novel classes suggest BaCon yields a more balanced feature space. Besides, our method significantly improves the performance of the minority classes without sacrificing the accuracy of the majorities.

We also report the performance of BaCon when the training set has a different imbalance ratio $\rho$ in Table 5. Note that we set the imbalance ratio of $\mathcal{D}^l$ and $\mathcal{D}^u$ to be the same, i.e., $\rho_l = \rho_u = \rho$. Empirical results with $\rho_l \neq \rho_u$ are in Appendix E. Notably, BaCon achieves the best *overall* accuracies on all settings, indicating the universal effectiveness across various scenarios, and the performance gain becomes more evident under extremely imbalanced data. Also, open-world recognition emphasizes *novel* class learning, where BaCon achieves a more notable performance gain.

| Strategy | Old | New | All |
|---|---|---|---|
| reg with $\boldsymbol{\pi}_{\mathcal{D}^l}$ | 76.4 | 13.2 | 44.8 |
| balance prior | 75.6 | 31.7 | 58.3 |
| cls k-means | 76.3 | 36.4 | 56.4 |
| con k-means | **81.7** | **42.3** | **62.0** |
| oracle | 85.0 | 44.2 | 64.6 |

(a) **Regularization Term.**

| Strategy | Old | New | All |
|---|---|---|---|
| baseline | 65.1 | 52.0 | 62.5 |
| vanilla | 66.3 | 62.4 | 65.5 |
| w/ debiasing | 67.1 | 64.1 | 66.5 |
| w/ sampling | 66.8 | 65.6 | 66.6 |
| w/ both | **67.4** | **66.5** | **67.2** |

(b) **Debiasing and Sampling.**

| Loss | Old | New | All |
|---|---|---|---|
| baseline | 65.1 | 52.0 | 62.5 |
| hard | 66.0 | 60.7 | 64.9 |
| soft | **67.4** | **66.5** | **67.6** |

(c) **Loss Design.**

| Interval $r$ | Old | New | All |
|---|---|---|---|
| 1 | **67.3** | 70.2 | **67.9** |
| 5 | 66.4 | **70.6** | 67.2 |
| 10 | 67.4 | 66.5 | 67.2 |
| 25 | 66.5 | 67.5 | 66.7 |
| 50 | 66.6 | 65.8 | 66.4 |

(d) **Re-estimate Interval $r$.**

| Function | Old | New | All |
|---|---|---|---|
| $L_1$ | 64.6 | 64.0 | 64.5 |
| $L_2$ | 63.2 | 60.9 | 62.7 |
| cosine | **67.8** | 64.1 | 67.0 |
| dot | 67.4 | **66.5** | 67.2 |

(e) **Similarity Function.**

| $k$ | Old | New | All |
|---|---|---|---|
| 0 | 65.8 | 67.8 | 66.2 |
| 0.25 | 66.1 | **69.2** | 66.7 |
| 0.5 | 67.4 | 66.5 | 67.2 |
| 1.0 | **67.5** | 65.0 | 67.0 |
| 1.5 | 66.9 | 66.2 | 66.8 |

(f) **Hyper-parameter $k$.**

Table 6: **BaCon ablation experiments.** For each ablation, we report test accuracy (%) of known, novel and all classes, denote as 'Old', 'New', and 'All'. Our default settings are marked in orange .

## 5.4 Analysis and Ablation Study

**The effectiveness of $\mathcal{L}_{reg}$** Recall in Section 4.1, we propose to regularize the predictions of the pseudo-labeling branch by the estimated train-set distribution. In Table 6a, we show the performance of the pseudo-labeling branch on ImageNet-100-LT with different estimation strategies. 'Oracle' denotes we use the true distribution $\boldsymbol{\pi}$ of $\mathcal{D}$ (unknown in practice) as the target distribution in Eq. 2, and it serves as an upper bound of the performance. Compared to previous works (ORCA and SimGCD) that use a balance prior, regularizing the predictions with oracle long-tailed distribution significantly improves the performance on both known and novel categories, showing the importance of the distribution estimation process. Meanwhile, the similar results achieved by our estimation strategy imply $\boldsymbol{\pi}_e$ could be a reliable proxy to $\boldsymbol{\pi}$. Furthermore, we investigate whether two alternative estimation strategies could help the pseudo-labeling branch: 1), only regularize known classes prediction with $\boldsymbol{\pi}_{\mathcal{D}^l}$ 2), perform k-means clustering on the feature of the pseudo-labeling branch, and they all results in inferior accuracy and could in turn deteriorate the contrastive learning process.

**The effectiveness of sampling & debiasing** In Section 4.2, we suggest adjusting the prediction logits according to the estimated distribution for debiasing and sampling unlabeled instances for re-balancing. We adopt step-by-step ablation experiments to the proposed approaches in Table 6b. 'Baseline' refers to not using the proposed pseudo-label-based contrastive loss, and it leads to inferior performance on novel classes since they only get supervision from the self-supervised contrastive loss. 'Vanilla' means incorporates the designed pseudo-label-based loss into the optimization objective with Eq. 7 without the debiasing and sampling module. It brings performance gain on novel classes due to the leverage of additional supervision in pseudo labels. Finally, we achieve higher test accuracy on both known and novel classes by combining the two techniques together.

**Definition of the pseudo-label based contrastive loss** In Section 4.3, we design a novel *soft* contrastive loss based on pseudo-labels to transfer the knowledge of $f_{cls}$ into $f_{con}$. As an opposite, we could also construct the loss in a *hard* manner where we formulate the positive pairs on top of the prediction class with the largest logit and further perform the supervised contrastive loss. Intuitively, the *hard* design discards the probability distribution information and is more susceptible to the potential false pseudo-labels, while the *soft* contrastive loss utilized in our method could help alleviate the influence of erroneous pseudo-labels. The empirical results also support the intuition, as shown in Table 6c, the proposed $\mathcal{L}_{CL}^s$ (soft) outperforms the supervised CL loss (hard) by a large margin. This phenomenon is also observed in knowledge distillation [25, 65] that transferring knowledge by using soft labels rather than one-hot predictions achieves better performance.

**Changing of metrics and hyper-parameters** The proposed soft CL loss introduces a pair-wise co-efficient to adjust the contribution of different samples to the anchor instance. We apply various similarity metrics on the pseudo-labels of $\boldsymbol{x}_i$ and $\boldsymbol{x}_p$ to calculate the *positiveness* score $w_{ip}$. The results are exhibited in Table 6e, we incorporate dot product in BaCon due to the superior performance

in practice. We also show the effect of hyper-parameters involved in BaCon. Table 6d shows the empirical results of changing estimate interval $r$. We observe similar accuracy when scaling $r$, hence we use a relatively large interval to reduce the computational overhead. Table 6f shows the classification accuracy when changing $k$ in Eq. 4.

**Feature Visualization** We adopt the widely-used T-distributed stochastic neighbor embedding (t-SNE) [58] for feature space visualization. We show the feature space of BaCon (ours) with DINO [6] pre-trained model (without fine-tuning) and GCD [60] in Figure 3, different colors represent data belonging to distinct categories. Compare to vanilla DINO (Figure 3a) and GCD fine-tuned model (Figure 3b), BaCon has more compact features for samples within the same category, suggesting better intra-class consistency, and also a larger margin between different classes.

## 6   Conclusion and Limitations

In this paper, we formulate a more realistic DA-GCD task that unifies long-tailed and open-world learning, and design a novel framework BaCon for DA-GCD to fight against data imbalance and classify open-set categories simultaneously. Extensive experiments on various datasets verify the effectiveness of the proposed method. There are nevertheless some limitations. First, our work focuses on the imbalanced open-world classification task, how to extend to object detection and instance segmentation is also worth exploring. Second, while the computation increment is not severe, how to further accelerate the proposed method will be explored in the future. We hope our work can promote the research on imbalanced open-set recognition in real-world applications.

## Acknowledgements

We thank the anonymous NeurIPS reviewers for providing us with valuable feedback that greatly improved the quality of this paper!

This work is supported by the Zhejiang Provincial Key RD Program of China (Grant No. 2021C01119), the National Natural Science Foundation of China (Grant No. U21B2004, 62106222), and the Core Technology Research Project of Foshan, Guangdong Province, China (1920001000498), the Natural Science Foundation of Zhejiang Province, China (Grant No. LZ23F020008) and the Zhejiang University-Angelalign Inc. R&D Center for Intelligent Healthcare.

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

# A Proof for Claim 1

*Proof.* We start with recalling some related notations and definitions. In Section 4.3, we design a novel soft contrastive loss by incorporating a pair-wise *positiveness* score $\boldsymbol{w}$ in Eq. 1:

$$\mathcal{L}_{\text{CL}}^{soft}(\mathcal{D}) = \frac{1}{|\mathcal{D}|} \sum_{i \in \mathcal{D}} \left[ \frac{1}{\sum_{j \in A(i)} w_{ij}} \sum_{j \in A(i)} -w_{ij} \cdot \log \frac{\exp(\boldsymbol{z}_i \cdot \boldsymbol{z}_j / \tau)}{\sum_{a \in A(i)} \exp(\boldsymbol{z}_i \cdot \boldsymbol{z}_a / \tau)} \right], \tag{8}$$

where $\mathcal{D}$ is the training set, $A(i)$ is the index of the set of instances involved in the same batch, $\boldsymbol{z}_i$ is the contrastive feature for instance $\boldsymbol{x}_i$, and $\tau$ is the temperature hyper-parameter.

For each instance $\boldsymbol{x}_i$, the soft contrastive loss $\mathcal{L}_{\text{CL}}^{soft}(\boldsymbol{x}_i)$ can be written as:

$$\mathcal{L}_{\text{CL}}^{soft}(\boldsymbol{x}_i) = C_i \sum_{j \in A(i)} \left[ -w_{ij} \cdot \log \frac{\exp(\boldsymbol{z}_i \cdot \boldsymbol{z}_j / \tau)}{\sum_{a \in A(i)} \exp(\boldsymbol{z}_i \cdot \boldsymbol{z}_a / \tau)} \right], \tag{9}$$

where $C_i = \frac{1}{\sum_{j \in A(i)} w_{ij}}$ is a constant value for fixed $\boldsymbol{w}$, then the contrastive logit $p_{ij}$ is defined as:

$$p_{ij} = \frac{\exp(\boldsymbol{z}_i \cdot \boldsymbol{z}_j / \tau)}{\sum_{a \in A(i)} \exp(\boldsymbol{z}_i \cdot \boldsymbol{z}_a / \tau)} \tag{10}$$

Then, we omit the constant $C$ in Eq. 9 and solve the optimization problem with the Lagrange multiplier. The problem is defined as follows:

$$\begin{cases} \text{Minimize:} \quad f(p_{i1}, \cdots, p_{in}) = -\sum_{k=1}^{n}(w_{ik} \cdot \log p_{ik}), \\[2mm] \text{Subject to:} \quad g(p_{i1}, \cdots, p_{in}) = \sum_{k=1}^{n} p_{ik} - 1 = 0, \end{cases} \tag{11}$$

where $n$ refers to the number of instances in the training batch, i.e., $n = |B| = |A(i)|$. The Lagrangian function of Eq. 11 and its corresponding partial derivatives are:

$$\mathcal{L}(p_{i1}, \cdots, p_{in}, \lambda) = -\sum_{k=1}^{n}(w_{ik} \cdot \log p_{ik}) + \lambda(\sum_{k=1}^{n} p_{ik} - 1), \tag{12}$$

$$\begin{cases} \frac{\partial \mathcal{L}}{\partial p_{ij}} = -\frac{w_{ij}}{p_{ij}} + \lambda = 0 \\[2mm] \frac{\partial \mathcal{L}}{\partial \lambda} = \sum_{k=1}^{n} p_{ij} - 1 = 0 \end{cases} \tag{13}$$

From Eq. 13, the optimal value of $p_{ij}$ is $p_{ij}^* = \frac{w_{ij}}{\sum_{k=1}^{n} w_{ik}}$, which concludes the proof for Claim 1.

Claim 1 indicates that minimizing Eq. 6 encourages the similarity of features between two samples to be proportional to the corresponding *positiveness* score. In this way, we effectively transfer knowledge from the pseudo-labeling branch to contrastive learning.

# B   Pseudo-code

We summarize the pipeline of self-**Ba**lanced **Co**-advice co**n**trastive framework (BaCon) in Algorithm 1, and the source code can be found here.

---

**Algorithm 1** The overall pipeline of BaCon.

---

**Input**: Train set $\mathcal{D} = \{\mathcal{D}^l \cup \mathcal{D}^u\}$, model parameter $f_{con}, f_{cls}$, train epoch $T$, warm-up epochs $w$, estimation interval $r$, sampling rate $\alpha, \beta$, similarity metric $\mathrm{Sim}\,(\cdot)$, hyper-parameter $p$ and $k$.
**Output**: Trained model parameter $\theta_T$.
**Initialize**: Load DINO [6] pre-trained parameters for the backbone in two branches and the classifier and projector is randomly initialized.

 **for** epoch $= 0, \cdots, T-1$ **do**
  **if** epoch$\%r = 0$ **then**
   Re-estimate the training set distribution and re-place $\boldsymbol{\pi}_e$ (Section 4.1);
  **end if**
  Regularize the predictions of the classifier with Eq. 2;
  Train $f_{cls}$ with Eq. 3;
  Compute the self-supervised CL loss $\mathcal{L}_{\mathrm{CL}}^u(\mathcal{D})$ and supervised CL loss $\mathcal{L}_{\mathrm{CL}}^s(\mathcal{D}^l)$ with Eq. 1;
  **if** $epoch \geq w$ **then**
   Debiasing pseudo-labels with Eq. 4;
   Obtain the sampled pseudo-labels $M(\mathcal{D}^u)$ with sampling rate in Eq. 5;
   Calculate the pair-wise *positiveness* score $\boldsymbol{w}$ with $\boldsymbol{w} = \mathrm{Sim}(\widetilde{\boldsymbol{p}}_i, \widetilde{\boldsymbol{p}}_j)$;
   Compute the *soft* CL loss $\mathcal{L}_{\mathrm{CL}}^{soft}(\mathcal{D}^l \cup M(\mathcal{D}^u))$ with Eq. 6;
   Train $f_{con}$ with Eq. 7.
  **else**
   Train $f_{con}$ with $\mathcal{L}_{\mathrm{CL}}^u(\mathcal{D}) + \gamma_1 \mathcal{L}_{\mathrm{CL}}^s(\mathcal{D}^l)$.
  **end if**
 **end for**

---

After training, we test the model via k-means clustering on the backbone feature of the contrastive-learning branch. The test strategy is summarized in Algorithm 2.

---

**Algorithm 2** The test stage strategy of BaCon.

---

**Input**: The balanced test set $\mathcal{D}^t = \{(\boldsymbol{x}_i, y_i)\} \in \mathcal{X} \times \mathcal{Y}$, fine-tuned backbone of the contrastive-learning branch $f_b$.
**Output**: Classification accuracy Acc.

 **for** $\boldsymbol{x}_i \in \mathcal{D}^t$ **do**
  Obtain the feature of $\boldsymbol{x}_i$ via $f_b(\boldsymbol{x}_i)$
 **end for**
 Perform k-means clustering on all the features of test samples with the cluster number $C = |\mathcal{D}^t|$;
 Calculate the optimal assignment between clusters and classes by Hungarian algorithm [32];
 Compute the test accuracy Acc based on the optimal assignment.

---

# C   Detailed Experimental Settings

In this section, we provide a comprehensive overview of the experimental setup, expanding upon the details presented in Section 5.1.

## C.1   Benchmark Datasets

In Section 5.3, we present our experimental findings based on four generic long-tailed image recognition datasets. CIFAR-10-LT and CIFAR-100-LT, which are first introduced by [10]. Both datasets are long-tail subsets sampled from the original CIFAR-10 and CIFAR-100. The imbalance ratio is defined as the ratio between the number of instances in the largest class and the smallest class. In our experiments, we default the imbalance ratio to 100 to reflect the performance disparities between classes. ImageNet-100-LT, proposed by [27], which comprises 12K images sampled from the ImageNet-100 dataset [57] using a Pareto distribution. The instance number of each class in the training set ranges from 1,280 to 5. We also incorporate the Places dataset [80], a large-scale scene-centric dataset, in our experiments. We utilize Places-LT [40], which consists of approximately 62.5K images sampled from the Places dataset using a Pareto distribution. The training set of Places-LT is ranging from 4,980 to 5. A summary of the dataset statistics is provided in Table 7.

Table 7: Statistics of datasets.

| Dataset | Long-tail type | Imbalance ratio $\rho$ | #class | #train images | #test images |
|---|---|---|---|---|---|
| CIFAR-10-LT | Exp | 100 | 10 | 11.2K | 10.0K |
| CIFAR-100-LT | Exp | 100 | 100 | 9.8K | 10.0K |
| ImageNet-100-LT | Pareto | 256 | 100 | 12.0K | 5.0K |
| Places-LT | Pareto | 996 | 365 | 62.5K | 36.5K |

## C.2   Construction of Training and Testing Sets

To construct the training set for distribution-agnostic generalized category discovery (DA-GCD), we first sub-sample $|\mathcal{Y}_k|$ classes from the entire long-tail datasets (introduced in Section C.1) to stimulate known classes and treat the rest as novel classes, then sub-sample 50% instances in each known class as $\mathcal{D}^l$ and concentrate others with all samples in novel classes to form $\mathcal{D}^u$. The default $|\mathcal{Y}_k|$ for CIFAR-10-LT, CIFAR-100-LT, ImageNet-100-LT, and Places-LT are 5, 80, 50, and 182 respectively. Our default training set of each dataset is summarized in Table 8. We also evaluate BaCon with different $\mathcal{Y}_k$, different labeled ratios $r_l$ (default $r_l = 50\%$), and different imbalance ratios of both labeled set and unlabeled set.

Table 8: The default setting of DA-GCD's training set.

| Dataset | Known classes $|\mathcal{Y}_k|$ | Novel classes $|\mathcal{Y}_n|$ | Labeled images $|\mathcal{D}^l|$ | Unlabeled images $|\mathcal{D}^u|$ |
|---|---|---|---|---|
| CIFAR-10-LT | 5 | 5 | 4.5K | 6.7K |
| CIFAR-100-LT | 80 | 20 | 3.8K | 6.0K |
| ImageNet-100-LT | 50 | 50 | 3.2K | 9.0K |
| Places-LT | 182 | 183 | 20.8K | 41.7K |

For the test set, we use a disjoint balanced set as a common practice in the field of long-tail learning [29, 31, 27]. It's worth noting that the test set contains *both* known and novel classes, and the model is required to classify test set instances to every single class.

## C.3   Implementation Details of BaCon

We implement all our techniques using PyTorch [47] and conduct the experiments using a RTX3090 GPU. Following GCD [60], we load a DINO [6] pre-trained ViT-B/16 [13] as the backbone for both

branches. The classification head and projector are randomly initialized. We only fine-tune the last block of the backbone, the classification head in the pseudo-labeling branch, and the projector in the contrastive-learning branch. We use the output of [CLS] token with a dimension of 768 as the backbone feature for an input image. The classifier is implemented with a single fully-connected layer, and we use the cosine classifier due to its competent empirical results in long-tail learning [29]. The projector is an MLP with an output feature dimension of 65536. We train with a batch size of 256, and an initial learning rate of 0.1 decayed with a cosine schedule. We train for 200 epochs on each dataset. The temperature $\tau$ is set to 1.0, the hyper-parameter $p$ and $k$ is set to 0.5, the sampling rates $\alpha = 0.8$, $\beta = 0.5$, and the re-estimate interval $r$ is 10 (epochs). We use dot product as the similarity metric to define the *positiveness* score $\boldsymbol{w}$.

# D   Introduction of Baseline Methods

In our paper, we formally define a more realistic task called distribution-agnostic generalized category discovery (DA-GCD) and design a novel framework BaCon for the challenge setting. To evaluate the effectiveness of BaCon, we compare the proposed method with SOTA techniques in two closely related fields: imbalanced semi-supervised learning (imbalanced SSL) and generalized category discovery (GCD). In this section, we provide a detailed introduction of baseline methods (including ABC [33] and DARP [31] in imbalanced SSL; GCD [60], ORCA [4], OpenCon [56], and SimGCD [70] in GCD).

Imbalanced SSL [31, 33, 69, 19] considers the scenario where the training set has a long-tailed distribution and only a small subset of data is labeled. DARP [31] suggests refining pseudo-labels to the long-tailed distribution of the training set, and further designing a method to estimate the unknown distribution based on the assumption that the confusion matrix for labeled data and the unlabeled part are almost the same. Unfortunately, their assumption does not always hold in practice since the model tends to overfit the labeled samples. ABC [33] introduces an auxiliary balanced classifier to mitigate the data imbalance issue. Nevertheless, they also require the class-wise data distribution which is agnostic (due to the presence of open-set class samples) in DA-GCD.

Generalized category discovery (GCD), formulated by [60], is a new-raised domain in open-set learning. GCD manages to jointly recognize *known* categories contained in the manually annotated subset as well as *novel* (open-set) classes which appeared in the unlabeled set. Note that GCD classified the open-set samples into fine-grained categories according to their visual concept, which is different from OOD detection [37, 24, 39]. The pioneering work GCD [60] proposes to leverage the pre-trained ViT and contrastive learning to learn a discriminative feature space and further obtain predictions by the proposed semi-k-means clustering strategy. ORCA [4] suggests balancing the learning rate of known and novel classes by an uncertainty adaptive margin mechanism. OpenCon [56] attempts to supervise the unlabeled instances by generating pseudo-positive pairs for closely aligned representations. But this paradigm could lead to another dilemma: the representation and pseudo-label are interdependent, which means, an inferior feature space (e.g., lack of intra-class consistency) could lead to false positive pairs, which deteriorate the learning of feature space in turn. A recent work SimGCD [70], argues that parametric classification could achieve better performance with elaborately designed pseudo-labeling techniques. However, the aforementioned methods in GCD are built on the class balance assumption, and we observe a consistent performance degradation when generalizing to DA-GCD scenarios.

# E  More Empirical Results

In this Section, we provide more empirical results on different datasets, and experimental settings to further verify the effectiveness of the proposed BaCon.

## E.1   Results with Different Ratio of Known and Novel Classes

Here we show the accuracy when changing the number of known (close-set) and novel (open-set) classes. A small $|\mathcal{Y}_k|$ means fewer categories have manually labeled samples (illustrated in Figure 1), making the learning process more difficult. The performance is reported in Table 9. It's revealed that BaCon achieves the best accuracies on all settings, and brings larger performance gain under more challenging settings, i.e., few known classes and a large number of novel categories (outperforms best baseline $\sim$7% overall accuracy on CIFAR-100-LT when $|\mathcal{Y}_k| = 20$). Note the default $|\mathcal{Y}_k|$ for CIFAR-10-LT, CIFAR-100-LT, ImageNet-100-LT, and Places-LT are 5, 80, 50, and 182 respectively.

Table 9: Test accuracy (%) on CIFAR-100-LT with different $|\mathcal{Y}_k|$.

| Methods | $|\mathcal{Y}_k| = 20$ | | | $|\mathcal{Y}_k| = 50$ | | | $|\mathcal{Y}_k| = 80$ | | |
|---|---|---|---|---|---|---|---|---|---|
| | Old | New | All | Old | New | All | Old | New | All |
| TRSSL[†] | **65.5** | 30.7 | 37.6 | 62.2 | 30.5 | 46.4 | 58.7 | 35.8 | 54.1 |
| ORCA[†] | 43.5 | 29.3 | 32.1 | 56.6 | 30.2 | 43.4 | 55.0 | 30.8 | 50.1 |
| SimGCD | 51.2 | 27.3 | 32.1 | 66.3 | 19.6 | 42.9 | 59.8 | 24.2 | 52.8 |
| GCD | 63.4 | 50.8 | 53.3 | 67.0 | 53.0 | 60.0 | 65.5 | 49.0 | 62.2 |
| OpenCon[†] | 60.3 | 37.2 | 41.8 | **69.0** | 33.3 | 51.2 | 64.2 | 40.9 | 59.6 |
| BaCon-O | 62.7 | 56.9 | 58.1 | 66.8 | 59.4 | 63.1 | 66.5 | **69.6** | 67.1 |
| BaCon-S | 64.2 | **59.2** | **60.2** | 67.9 | **60.2** | **64.0** | 67.4 | 66.5 | **67.2** |

## E.2 Results on Few-annotated Scenarios

In Table 10, we also conduct experiments under few-annotated scenarios, i.e., $r_l = \{10\%, 30\%, 50\%\}$. The annotation ratio $r_l$ is computed by $r_l = \frac{\#\text{labeled samples in a known class}}{\#\text{all samples in a known class}}$, since we only have access to manually annotated images from known categories (illustrated in Figure 1c), and all novel class instances are unlabeled. Note that we set $r_l = 50\%$ in default in the main paper.

Table 10: Test accuracy (%) with different annotation ratio $r_l$.

| Methods | $r_l = 10\%$ | | | $r_l = 30\%$ | | | $r_l = 50\%$ | | |
|---|---|---|---|---|---|---|---|---|---|
| | Old | New | All | Old | New | All | Old | New | All |
| TRSSL[†] | 53.7 | 32.6 | 49.5 | 56.0 | 32.6 | 51.4 | 58.7 | 35.8 | 54.1 |
| ORCA[†] | 43.8 | 41.3 | 43.3 | 51.2 | 32.2 | 47.4 | 55.0 | 30.8 | 50.1 |
| SimGCD | 51.2 | 17.6 | 44.5 | 55.7 | 25.6 | 49.7 | 59.8 | 24.2 | 52.8 |
| GCD | 61.7 | 54.9 | 60.4 | 63.1 | 58.8 | 62.2 | 65.5 | 49.0 | 62.2 |
| OpenCon[†] | 52.1 | 40.9 | 49.9 | 61.8 | 43.9 | 58.2 | 64.2 | 40.9 | 59.6 |
| BaCon-O | 63.2 | 58.8 | 62.3 | 64.9 | 58.8 | 62.3 | 66.5 | **69.6** | 67.1 |
| BaCon-S | **64.0** | **60.5** | **63.3** | **65.6** | **65.6** | **65.6** | **67.4** | 66.5 | **67.2** |

### E.3 Results with Different Imbalance Ratio on Labeled and Unlabeled Set

In Table 5, we report the performance of BaCon and baseline methods in GCD when the training set has a different imbalance ratio $\rho$. Now, we further set the imbalance ratio of labeled data and unlabeled data to be unequal ($\rho_l \neq \rho_u$), which is common in practice. For example, in medical image processing, the labeled data and unlabeled parts may come from hospitals in different countries. When it comes to autonomous driving, the occurrence frequency of stones or pedestrians may slightly change when the training data is from city areas or mountain roads.

The results are summarized in Table 11. We set the imbalance ratio of labeled data to 100 and vary the value of $\rho_u$ across different settings, namely $\{20, 50, 100, 150\}$. It's observed that the proposed BaCon outperforms all baseline methods by a large margin, indicating that BaCon is robust to the shift of imbalance ratio between labeled and unlabeled samples.

Table 11: Test accuracy (%) on CIFAR-100 ($\rho_l = 100$) with different $\rho_u$

| Methods | $\rho_u = 20$ | | | $\rho_u = 50$ | | | $\rho_u = 100$ | | | $\rho_u = 150$ | | |
|---|---|---|---|---|---|---|---|---|---|---|---|---|
| | Old | New | All | Old | New | All | Old | New | All | Old | New | All |
| TRSSL[†] | 63.1 | 38.8 | 58.2 | 57.2 | 44.0 | 54.6 | 58.7 | 35.8 | 54.1 | 55.5 | 39.0 | 52.2 |
| ORCA[†] | 56.4 | 45.5 | 54.2 | 56.9 | 42.1 | 53.9 | 55.0 | 30.8 | 50.1 | 56.9 | 31.9 | 51.8 |
| SimGCD | 61.1 | 27.9 | 54.4 | 59.2 | 29.2 | 53.2 | 59.8 | 24.2 | 52.8 | 59.6 | 24.7 | 52.6 |
| GCD | 65.6 | 56.1 | 63.7 | 65.5 | 54.1 | 63.2 | 65.5 | 49.0 | 62.2 | 64.5 | 50.9 | 61.8 |
| OpenCon[†] | 66.8 | 48.8 | 63.2 | 64.7 | 51.7 | 62.1 | 64.2 | 40.9 | 59.6 | 64.1 | 40.2 | 59.3 |
| BaCon-O | 67.1 | 63.4 | 66.4 | 67.0 | 61.0 | 65.8 | 66.5 | **69.6** | 67.1 | 65.3 | 54.4 | 63.1 |
| BaCon-S | **69.2** | **66.2** | **68.6** | **68.7** | **65.3** | **68.0** | **67.4** | 66.5 | **67.2** | **67.8** | **64.2** | **67.1** |

