# OpenReview forum: "Towards Distribution-Agnostic Generalized Category Discovery"
_NeurIPS.cc/2023/Conference — NeurIPS 2023 poster_

### Official Review · Reviewer_7Jpa · 2023-07-04

**Soundness:** 2 fair
**Presentation:** 2 fair
**Contribution:** 2 fair
**Rating:** 6
**Confidence:** 3

**Summary:**

This paper presents a novel real-world driven problem setting named long-tailed open-world classification (LT-OPC) setting, where a model should predict both closed-set and open-set classes with long-tailed distribution. To address such challenges, a Self-Balanced Co-Advice contrastive framework (BaCon) is proposed, where both contrastive learning branch and pseudo-labeling branch provide supervision for the LT-OPC task. First, the contrastive learning branch estimates the training distribution of a data to regularize the pseudo-labeling branch. In the pseudo-labeling phase, a debiasing and sampling phase is designed to resolve the data imbalance and insufficient supervision for the novel classes. In final, pseudo-label branch further provides supervision to the contrastive learning branch, in the form of  pseudo-label-based soft contrastive learning objective. A series of experiments are carried out under the proposed setting and the effectiveness of each of the proposed component is studied.

**Strengths:**

- The proposed LT-OPC setting with long-tailed distribution data considers a more practical scenario in real world, compared to the traditional semi-supervised learning and generalized category discovery setting.

- It seems that the contrastive learning strategy and pseudo-labeling strategy with interaction between them are well designed to deal with the proposed LT-OPC setting. The ablation results from the Table 6 can support its design.

- Experimentally, the proposed BaCon framework shows competitive results from the baseline methods in imbalanced SSL and generalized category discovery setting.

**Weaknesses:**

- It is unclear that the proposed problem setting of LT-OPC is novel. There have been some published works that tried to combine semi-supervised learning and generalized category discovery setting in the context of open-world semi-supervised learning [C1, C2], but any discussions or comparisons with them are not provided in this paper. To my view, the only difference is whether the training data has long-tailed distributions or not, which can weaken the task novelty of this work.

- It is unclear that the comparisons with imbalanced SSL methods are fair. To my understanding, the model trained with the proposed BaCon objective gets supervision for the novel classes from the unlabeled data with the pseudo-labeling term, while the baseline SSL methods do not. In that sense, it is natural that the proposed method shows higher accuracy in ‘New’ category classes. Any clarification or opinion from the authors is expected. In addition, for the imbalanced SSL baselines, both DARP and ABC do not contain contrastive branch. To make the comparison more fair, considering additional SSL baselines including contrastive learning branch is necessary [C3, C4].

- For the training the baseline methods, is the training setup identical to the procedure in the proposed BaCon method? Training BaCon model relies on transformer architecture with strong pretraining method of DINO. It is unclear that the training protocol for the other baseline methods is the same with BaCon for fair comparison.

- Due to combining two different mechanisms, which are pseudo-labeling and contrastive learning, many hyper-parameters are introduced. How those values can be determined in a new dataset without simply tuning the real test set of each benchmark? Considering a real-world scenario, the prior for the novel class categories can be prohibited. To simplify the question, can a model tuned with a dataset including both known classes and novel classes (e.g., validation data) generalize to a new dataset including 'unseen' novel classes (e.g., test data)? Detailed discussion on such assumption would be appreceated.

---

References

[C1] Rizve, Mamshad Nayeem, et al. "Openldn: Learning to discover novel classes for open-world semi-supervised learning." *European Conference on Computer Vision*. Cham: Springer Nature Switzerland, 2022.

[C2] Rizve, Mamshad Nayeem, Navid Kardan, and Mubarak Shah. "Towards realistic semi-supervised learning." *European Conference on Computer Vision*. Cham: Springer Nature Switzerland, 2022.

[C3] Park, Jongjin, et al. "Opencos: Contrastive semi-supervised learning for handling open-set unlabeled data." *European Conference on Computer Vision*. Cham: Springer Nature Switzerland, 2022.

[C4] Zheng, Mingkai, et al. "Simmatch: Semi-supervised learning with similarity matching." *Proceedings of the IEEE/CVF Conference on Computer Vision and Pattern Recognition*. 2022.

**Questions:**

- The notation of $\mathcal{Y}_{n}$ in line 113 is used without definition.

- What exact type of loss function is utilized for the pseudo-labeling loss $\mathcal{L}_{u}$ from the equation 3?

- It can be found that the codes are attached, but it is unclear the codes can be executable. If you decided to include the codes in the submission, making the codes more understandable would be appreciated.

**Limitations:**

Brief limitation of the work can be found, but potential negative societal impact has not been addressed.

---

> ### Author Rebuttal · Authors · 2023-08-10
>
> > Novelty of LT-OPC.
>
> Please kindly refer to 'The Novelty of the Proposed Setting' in the 'General Response' column at the top.
>
> > Discussions or comparisons with [C1, C2] are not provided.
>
> Thank you for the reminder! We have migrated the two methods [C1, C2] you mentioned to the proposed LT-OPC scenario, and we will include them as baseline methods in the final version. The results are shown in the rebuttal PDF file as Table A and Table B. It is worth noting that for a fair comparison, we have replaced the backbone model of both methods with the more powerful DINO pre-trained ViT, which is consistent with the backbone model of our method.  Moreover, we will provide introductions to these two methods in the 'Related Works' section.
>
> In addition, the two methods share similarities with the GCD scenario, which involves the consideration of open-set class samples in unlabeled data within a semi-supervised setting. During testing, the model is required to classify both close-set and open-set samples. In Table 1 and lines 28 to line 44 of the main text, we provide a detailed comparison between the proposed LT-OPC and relevant scenarios such as GCD. We will incorporate the comparisons with these two methods in the respective location.
>
> > Model trained with BaCon gets supervision for the novel classes from the pseudo-labeling term, while the baseline SSL methods do not.
>
> Firstly, we would like to clarify that the dataset used for training the baseline SSL methods and our method are exactly the same, as shown in Figure 1(a) in the main text. The training set consists of labeled close-set samples, and unlabeled close-set and open-set samples.
>
> Secondly, we would like to claim that **the baseline SSL methods also utilize ‘the pseudo-labeling term’ during training** to obtain supervision for the unlabeled samples, which is the same as our method. Specifically, their methods first generate pseudo-labels for each sample, then filter out pseudo-labels with confidence scores lower than a pre-set threshold. The remaining pseudo-labels are then used as supervision for training. On the other hand, our method trains a pseudo-labeling branch to generate pseudo-labels, which in turn guides contrastive learning. In this process, **no additional supervision**, such as the true labels of the 'new' categories samples, is used. Therefore, we assume that our comparison with the baseline SSL methods is fair.
>
> > Considering additional SSL baselines including contrastive learning branch is necessary [C3, C4].
>
> Thank you for your suggestion! We conducted experiments on the two methods, Opencos [C3] and Simmatch [C4], and reported the results in Table E. To ensure fairness, we re-implemented both methods using DINO pre-trained ViT as the backbone model. We will include the results in the final version.
>
> > It is unclear that the training protocol for the baseline methods is the same with BaCon.
>
> Please kindly refer to 'The Fairness of the Experiments' in the 'General Response' column at the top.
>
> > How hyper-parameters can be determined in a new dataset without simply tuning the real test set?
>
> To verify whether our method can be extended to 'unseen' novel classes, we conducted experiments on ImageNet. Specifically, we first randomly sample 50 categories as 'known' classes and keep these 50 classes consistent across all experiments. Then, we **non-repetitively** sample 6x50 categories from the remaining 950 categories, resulting in six groups of classes named 'novel_val' and 'novel_test_A/B/C/D/E'. Next, we select the optimal hyperparameters based on the performance on the {'known' + 'novel_val'} dataset. **By Keeping all the hyperparameters fixed**, we further train and evaluate the model on the five different test datasets.  The experimental results shown in Table G, revealed that the hyperparameters performing best on the validation set achieved similar performance on different 'unseen' novel classes. This indicates that our method exhibits good generalization, implying its effectiveness in handling 'unseen' novel classes. We will report the experiment results in the final version.
>
> > y_n in line 113 is used without definition.
>
> We would like to thank the reviewer for a very detailed review. $\mathcal{Y}_n$ represents the set of labels for novel classes, which corresponds to the total number of categories in the open-set data. We will include this clarification in Section 3.1.
>
> > What exact type of loss function is utilized for L_u from Eq. 3?
>
> In Equation 3, the term $\mathcal{L}_u$ can be any unsupervised loss function used in classifier-based GCD methods. In our experiments, we choose two different $\mathcal{L}_u$ to demonstrate the effectiveness of our method. These are the cross-pseudo supervision loss from SimGCD (BaCon-S) and the pairwise objective from ORCA (BaCon-O).  Specifically, the cross-pseudo supervision loss is a loss function similar to the idea of self-distillation and SwAV [1]. It uses the predictions of two views (images augmented differently) of the same image as pseudo-labels for each other and calculates the cross-entropy loss. On the other hand, the pairwise objective in ORCA encourages intra-class consistency by pulling together samples with similar representations.  From the experimental results in Section 5, it's observed that our method can adapt to different $\mathcal{L}_u$ and achieve significant improvements, demonstrating the effectiveness and versatility of the proposed approach.
>
> > It is unclear the codes can be executable.
>
> We apologize for the inconvenience. Some of the code was not successfully synchronized from my GitHub repository to the anonymous repository. We have now resolved this issue and have attached a readme file as an explanation. Thanks for the reminder!
>
> [1] Caron M, et al. Unsupervised learning of visual features by contrasting cluster assignments. NIPS, 2020.

---

> > ### Comment · Reviewer_7Jpa · 2023-08-14
> > **Post-rebuttal comment**
> >
> > I acknowledge the authors' effort in the rebuttal. After carefully checking comments from other reviewers and the rebuttal, I would like to provide follow-up comments and questions.
> >
> > > Novelty of LT-OPC
> >
> > I am not fully convinced about the task novelty. While considering imbalance in data is practical, but it just feels to me a straightforward extension of GCD settings to long-tailed distributions. In addition, one can think that existing GCD methods such as SimGCD can cover such long-tailed distributions with clustering-based re-balancing techniques upon both closed and open categories, similar to the proposed work.
> >
> >
> > > Model trained with BaCon gets supervision for the novel classes from the pseudo-labeling term, while the baseline SSL methods do not.
> >
> > This comment was not about either dataset itself or pseudo-labeling branch, but the category space for the supervision between the proposed work and the other SSL works. To my understanding, the estimated distribution $\pi_c$ from k-means clustering (Eqn 2 of the main paper) includes both closed and open-set classes as category space. Due to this, the loss terms from equations 2, 6, and 7 can contain supervision signals for open-set classes. On the other hand, SSL methods with only pseudo-labeling branch get supervision signals only from closed-set categories. In that sense, I think it is natural that the proposed method shows much higher accuracy on novel categories. If I am wrong, please let me know.
> >
> >
> > For the other comments in rebuttal, additional experiments on fair settings (i.e.,  pretraining and backbone) and tuning hyper-parameters on 'valid' novel classes are encouraging. In summary, I would raise my score recommendation by one (3 -> 4) for its promising results.

---

> > > ### Author Response · Authors · 2023-08-15
> > > **Further Reply to Reviewer 7Jpa (1/2)**
> > >
> > > > I am not fully convinced about the task novelty. While considering imbalance in data is practical, but it just feels to me a straightforward extension of GCD settings to long-tailed distributions.
> > >
> > > Indeed, it could be seen as a straightforward extension of the GCD setting, but it's a much more challenging scenario to consider long-tail distribution in an open world. More importantly, it cannot be effectively addressed by existing approaches in long-tail learning because: (1) most supervised or semi-supervised long-tail learning methods [1, 2, 3] require the prior training set (long-tailed) distribution to tackle the data imbalance issue, and it prevents them from being directly extended to resolve the proposed LT-OPC task, since the prior dataset distribution is unavailable in LT-OPC (line 56 - line 58). (2) though several self-supervised methods [4, 5] could alleviate the imbalance issue without knowing the dataset distribution, they discard all the accessible label information and bring marginal accuracy gain. As shown in Table C (please kindly refer to the rebuttal PDF file), we combine two best-performing baseline methods GCD and SimGCD with BCL [5], which is one of the latest self-supervised long-tail learning methods. It's observed that BCL cannot effectively tackle the data imbalance issue, while the proposed BaCon improves both accuracy and balancedness by a large margin via dynamic distribution estimation (Section 4.1) and the self-balanced knowledge transfer module (Section 4.2).
> > >
> > > > In addition, one can think that existing GCD methods such as SimGCD can cover such long-tailed distributions with clustering-based re-balancing techniques upon both closed and open categories, similar to the proposed work.
> > >
> > > In fact, we initially thought of modifying methods such as SimGCD to generalize it to our proposed LT-OPC scenario. However, our experimental results indicate that improving these methods via long-tail learning approaches cannot achieve satisfying performance. Specifically, we replace the uniform distribution prior (on both close and open categories) used in SimGCD to the **oracle distribution** of the long-tailed dataset in LT-OPC, which means the distribution of both labeled and unlabeled data, and it's not available in practice. Then, we evaluate the performance of SimGCD equipment with the oracle distribution, which can be regarded as the upper-bound performance for SimGCD, and the results are summarized below (in the second box).
> > >
> > > It's observed that the performance of SimGCD still has an inferior performance on LT-OPC, even though we have provided the oracle distribution to it for regularization, while the proposed BaCon outperforms them by a large margin. It indicates that simply modifying methods in GCD with re-balancing techniques cannot perform well in LT-OPC. While BaCon outperforms baselines by the designing of the dual branch structure which works collaboratively to provide interactive supervision and achieve self-rebalancing, also, the contrastive-learning branch takes advantage of the 'self-balanced knowledge transfer' module, which helps to learn a balanced and reasonable feature space.
> > >
> > > > This comment was not about either dataset itself or pseudo-labeling branch, but the category space for the supervision between the proposed work and the other SSL works. To my understanding, the estimated distribution from k-means clustering (Eqn 2 of the main paper) includes both closed and open-set classes as category space. Due to this, the loss terms from equations 2, 6, and 7 can contain supervision signals for open-set classes. On the other hand, SSL methods with only pseudo-labeling branch get supervision signals only from closed-set categories. In that sense, I think it is natural that the proposed method shows much higher accuracy on novel categories. If I am wrong, please let me know.
> > >
> > > Sorry for the misunderstanding about the question. Indeed, it's very crucial for SSL methods to get supervision signals from open-set samples. And we would like to clarify that we have provided the imbalanced SSL methods (i.e., ABC and DARP) with the oracle distribution in experiments, otherwise, they would have nearly zero accuracies on novel classes. We will add the description in the final version to avoid ambiguity.
> > >
> > >
> > >
> > > [1] Menon, Aditya Krishna, et al. "Long-tail learning via logit adjustment." ICLR 2021.
> > >
> > > [2] Kim, et al. "Distribution aligning refinery of pseudo-label for imbalanced semi-supervised learning." NIPS 2020.
> > >
> > > [3] Lee, et al. "ABC: Auxiliary balanced classifier for class-imbalanced semi-supervised learning." NIPS 2021.
> > >
> > > [4] Jiang, Ziyu, et al. "Self-damaging contrastive learning." ICML, 2021.
> > >
> > > [5] Zhou, Zhihan, et al. "Contrastive Learning with Boosted Memorization." ICML, 2022.

---

> > > > ### Author Response · Authors · 2023-08-15
> > > > **Further Reply to Reviewer 7Jpa (2/2)**
> > > >
> > > > | Method        | Many$\uparrow$ | Med$\uparrow$ | Few$\uparrow$ | Std$\downarrow$ | All$\uparrow$ | Many$\uparrow$ | Med$\uparrow$ | Few$\uparrow$ | Std$\downarrow$ | All$\uparrow$ | Overall  |
> > > > | ------------- | :------------: | ------------- | :-----------: | :-------------: | :-----------: | :------------: | :-----------: | :-----------: | :-------------: | :-----------: | :------: |
> > > > | SimGCD        |      72.8      | 71.1          |     34.6      |      17.6       |     59.8      |      32.9      |     25.8      |     15.5      |       7.1       |     24.8      |   52.8   |
> > > > | SimGCD-Oracle |      69.4      | **72.9**      |     45.6      |      12.1       |     62.9      |      47.0      |     43.9      |     36.8      |       4.3       |     42.7      |   58.9   |
> > > > | BaCon-O       |      72.9      | 64.7          |   **62.0**    |     **4.6**     |     66.5      |      72.6      |     70.6      |   **65.3**    |     **3.1**     |     69.6      |   67.1   |
> > > > | BaCon-S       |    **73.5**    | 65.6          |     61.2      |       5.1       |   **66.8**    |    **74.7**    |   **73.1**    |     64.5      |       4.5       |   **71.0**    | **67.6** |

---

> > > > ### Comment · Reviewer_7Jpa · 2023-08-18
> > > >
> > > > The provided responses resolve my main concern and questions. I find that the key task novelty comes from how to address the open-world distributions without any given prior, which cannot be simply undertaken with previously established generalized category discovery and semi-supervised learning literature. From multiple rounds of discussions with authors, this point is made clear and I believe the value of this work has significantly improved. I hope such points can be reflected on the revised version.
> > > >
> > > > Accordingly, I am pleased to recommend the rating to 6 (weak accept), finally. Once again, I appreciate the thorough responses from the authors in addressing my concerns.

---

> > > > > ### Author Response · Authors · 2023-08-18
> > > > > **Further Reply to Reviewer 7Jpa**
> > > > >
> > > > > As you mentioned, our main task novelty lies in effectively addressing the long-tail problem in an open-world setting without prior knowledge of the underlying distribution. We fully agree that we should emphasize and highlight this novelty in the final version, and we will further elaborate on this point in the revised paper.
> > > > >
> > > > > We sincerely appreciate your recognition of our work! We are extremely grateful for your thorough review of our work, as it has greatly helped us improve. Thank you once again for your time, and sincerely wish you all the best!

---

### Official Review · Reviewer_B9Lc · 2023-07-07

**Soundness:** 3 good
**Presentation:** 2 fair
**Contribution:** 3 good
**Rating:** 6
**Confidence:** 3

**Summary:**

The paper proposes a combination of contrastive learning and semi-supervised learning approach to label images in open-world classification. The classes are assumed to be long-tailed, as is the case in real-world cases. Not all the classes have a labeled example, but the number of classes are assumed to be known beforehand.

The proposed method follows a "two branch structure", which one branch doing contrastive learning, and another doing pseudo-labeling.

**Strengths:**

1. A highly relevant problem statement for real-world application of image classification
2. Thorough comparison against related works, both in conceptual differences and in evaluation
3. Soft contrastive loss is an interesting idea


**Weaknesses:**

Some of the algorithmic steps are not clearly described. Questions listed below.


**Questions:**

1. Line 146: What do you mean sample number n?
2. Line 146: What does the symbol (n^c) mean? Are you doing "n choose c" or "n raised to c"?
3. How is Hungarian optimal assignment used in the context of the problem? How do you define "cost" in the adjacency matrix? The citation provided does not explain this.
4. What is the "align" operation in equation 2?
5. Why does equation 4 lead to a "post-hoc logits adjustment"? It is better to assume the reader has not read citation 39 and explain the concept briefly
6. How is equation 5 leading to "we prioritize selecting samples with higher prediction confidence in each class"? Line 179.
7. Line 183. How do you mitigate imbalance issue with this sample process? Even the ground truth labels are highly skewed.
8. What is A(i) in equation 1 and 6?


**Limitations:**

Good paper overall. I would like to see a clearer explanation of methods as listed above.
The experiments where the number of classes are unknown are perhaps of higher importance, but they have been pushed to Appendix. It would be good to include a summary in the main paper.

---

> ### Author Rebuttal · Authors · 2023-08-10
>
> > Line 146: What do you mean sample number n?
>
> The sample number **n** is a vector with size $C$, where the $i$-th element represents the number of samples in $i$-th cluster (line 144- line 145).
>
> > Line 146: What does the symbol (n^c) mean? Are you doing "n choose c" or "n raised to c"?
>
> $n^c$ refers to the $c$-th element in vector $\boldsymbol{n}$, which is the number of samples in the $c$-th cluster. We realize that there may be ambiguity here, so we will change $n^c$ to $n[c]$ in the final version. Thank you for your reminder.
>
> > How is Hungarian optimal assignment used in the context of the problem? How do you define "cost" in the adjacency matrix? The citation provided does not explain this.
>
> Assuming we want to calculate the accuracy of a classification task with a number of categories $C$, the cost matrix $W \in \mathbb{R} ^ {C \cdot C}$. In this case, $-w_{ij}$ is the number of samples with the highest logits value in the $i$-th dimension of the model classifier (i.e., predict to dimension $i$), and the true label should be the $j$-th class of samples. We take the minus sign because the Hungarian optimal assignment algorithm will **minimize** the cost, and we want to find an optimal assignment that **maximizes** the test accuracy. After defining such a cost matrix, we can use the Hungarian algorithm to obtain the optimal assignment, under which the model has the maximum test accuracy. We will add the explanation in the appendix in the final version.
>
> > What is the "align" operation in equation 2?
>
> The 'align' operation means finding the correspondence between the clusters and categories using the method described from line 146 to line 149. In a word, we first map the 'known' classes via the Hungarian algorithm performed on labeled samples, then we sort the remaining clusters by the cluster size and assign them sequentially to the novel classes.
>
> > Why does equation 4 lead to a "post-hoc logits adjustment"? It is better to assume the reader has not read citation 39 and explain the concept briefly
>
> Thanks for the advice! Equation 4 implements the Logits Adjustment [1] method, which is one of the SOTA methods in supervised long-tail learning, its core idea is to adjust the output logits at test time to mitigate the issue of data distribution differences between testing (balanced) and training (long-tailed). In other words, the adjustment term $-k \cdot \pi_e$ introduces a label-relevant offset, which amplifies the tail class probability according to the data distribution during training, leading to an unbiased probability estimation. We will add a brief explanation to Section 4.2.
>
> > How is equation 5 leading to "we prioritize selecting samples with higher prediction confidence in each class"? Line 179.
>
> In fact, we only use Equation 5 to determine the number of samples that should be sampled for each category. After determining the number of samples, we then rank each category of samples in descending order of confidence and sequentially select the sample with the highest confidence according to the number of samples needed.
>
> > Line 183. How do you mitigate imbalance issue with this sample process? Even the ground truth labels are highly skewed.
>
> The long-tailed distribution of the dataset can indeed bring difficulties to re-balancing. However, since the sampling rate we set is inversely proportional to the number of samples in that category, we can assume that the number of samples sampled in each category is roughly the same, which can still alleviate the problem of long-tailed distribution, i.e., the data for contrastive learning has a smaller imbalance ratio after incorporating the sampled images.
>
> > What is A(i) in equation 1 and 6?
>
> $A(i)$ is the set of indices of all other features (both positive and negative pairs) in contrastive learning. Take SimCLR [2] for instance, assume we train the model with a batch size B, and all samples are augmented twice in different augmentations ($2B$ views), then feed into the model to obtain $2B$ features $\left\\{z_i\right\\}_{i=1}^{2B}$. For feature $z_i$, $A(i)$ refers to the set of indices of all other $(2B-1)$ features in the training batch. We will add the clarification in the main text.
>
> > The experiments where the number of classes are unknown are perhaps of higher importance, but they have been pushed to Appendix. It would be good to include a summary in the main paper.
>
> Thanks for the advice! We will add a summary of the experiments in the Appendix in Section 5.1.
>
>
>
> [1] Menon, Aditya Krishna, et al. "Long-tail learning via logit adjustment." ICLR, 2021.
>
> [2] Ting Chen, et al. A Simple Framework for Contrastive Learning of Visual Representations. ICML, 2020.

---

> > ### Comment · Reviewer_B9Lc · 2023-08-18
> > **Acknowledging Response**
> >
> > Dear Authors,
> >
> > Thank you for your thoughtful responses. The questions I raised have been answered adequately.

---

> > > ### Author Response · Authors · 2023-08-18
> > > **Further Reply to Reviewer B9Lc**
> > >
> > > Thank you very much for your valuable questions and suggestions! We sincerely appreciate them and we will make comprehensive revisions to our work based on your comments in order to further improve the quality of our work.
> > >
> > > Thanks again and best wishes!

---

### Official Review · Reviewer_udGd · 2023-07-07

**Soundness:** 2 fair
**Presentation:** 2 fair
**Contribution:** 2 fair
**Rating:** 4
**Confidence:** 3

**Summary:**

This paper tackles long-tailed open-world classification problem. It handles data imbalance in the long-tailed problem, and it also needs to handle the closed-world/open-world classifications.  For open-world classifications, there are only unlabeled data. Moreover, it assumes that unlabeled data can come from the "closed-world"classes as well.

It proposes a framework, called "Self-Balanced Co-Advice contrastive framework (BaCon)", to tackle data imbalance and handle the open-set classifications. The framework consists of a contrastive-learning branch and a pseudo-labeling branch, and the two branches also work collaboratively. Experiments are performed on image classification datasets.

**Strengths:**

The proposed problem of "long-tailed open-world classification" is valid. Both "long-tailed" and "open-world" are popular and important problems in vision.  This paper combines problems and approaches from a couple of previous works. The proposed approaches may work.

**Weaknesses:**

The proposed problem setting is not very novel. For example, the previous work of "Large-scale long-tailed recognition in an open world" (reference [36]) handles long-tailed problems in an open world. Also the paper "Generalized Category Discovery" (reference [52]) proposes "the unlabelled images may come from labelled classes or from novel ones."

The proposed approach is not novel. For example, the contrastive learning idea has been used in the "Generalized Category Discovery" (reference [52]).  Or the key insights need to be clearly stated.

**Questions:**

The experiments are done on CIFAR, ImageNet, and Places datasets. Are there any experiments on other fine-grained long-tailed datasets?

**Limitations:**

yes

---

> ### Author Rebuttal · Authors · 2023-08-10
>
> > The proposed problem setting is not very novel. For example, the previous work of "Large-scale long-tailed recognition in an open world" (reference [36]) handles long-tailed problems in an open world. Also the paper "Generalized Category Discovery" (reference [52]) proposes "the unlabelled images may come from labelled classes or from novel ones."
>
> GCD [1] and OLTR [2], are indeed related to the proposed LT-OPC setting, but we would like to clarify that the proposed problem setting still has significant differences from them and introduces unique challenges in this scenario. **Compared to GCD,** they only focus on the problem of the open world but ignore the data imbalance issue in the real world, i.e., their method falls in the lower right corner of Figure 1 (b); while our setting aims to solve both the problems of the open-world and data imbalance, corresponding to the upper right corner of Figure 1 (b). Moreover, since we do not know the prior distribution of the dataset in open-world scenarios, we cannot simply transfer existing long-tail learning methods to solve the problem of data imbalance in LT-OPC scenarios, making it more challenging. **Compared to OLTR,** though we both study the long-tail problem in open-world scenarios, our method pushes 'one step further'. That is, not only learn a model to detect the open-set samples, but it also requires the model to classify them according to their semantics. On the other, OLTR considers a fully-supervised scenario that only utilizes manually annotated samples (lower left in Figure 1 (c)); while we also make use of unlabeled (close-set and open-set) data (upper right and lower right in Figure 1 (c)), which makes our approach more powerful and generalizable.
>
>
>
> > The proposed approach is not novel. For example, the contrastive learning idea has been used in the "Generalized Category Discovery" (reference [52]). Or the key insights need to be clearly stated.
>
> Our **key insight** is methods like GCD [1] lack effective supervision of novel categories, since they only perform self-supervised contrastive loss on samples in novel classes, which leads to inferior performance. Meanwhile, GCD overlooks the long-tail distribution in the real world. '***How to simultaneously tackle the issue of data imbalance and provide adequate supervision for open-set classes***' motivates our proposed method BaCon, where the **key technical novelty** is the design of a dual branch structure and the co-advice mechanism which lets the two branches work collaboratively to provide interactive supervision to tackle the imbalanced recognition task under open-world scenarios. To be specific, the contrastive-learning branch provides distribution estimation to regularize the prediction results of the linear classifier for better pseudo-labeling. On the other, the generated pseudo-labels are sampled and debiased to re-balance and provide additional supervision for the contrastive-learning branch. To fuse the knowledge of the two branches and learn a better feature space, we design a novel pseudo-label-based contrastive loss that clusters samples based on their *positiveness* scores.
>
> Thanks for the advice! We will highlight our key insight and key technical novelty in the final version.
>
> > The experiments are done on CIFAR, ImageNet, and Places datasets. Are there any experiments on other fine-grained long-tailed datasets?
>
> In addition to conducting experiments on CIFAR-10-LT, CIFAR-100-LT, ImageNet-100-LT, and Places-365-LT as described in the Experiments section of the main text, we also present the experimental results of our method on ImageNet-1k-LT and iNaturalist [3] in Appendix E.5, and the iNaturalist dataset is highly imbalanced which is consists of 8,142 categories. Furthermore, we also compare the performance of our method with other baseline methods on the Herbarium-19 [4] dataset in Table F (please kindly refer to the rebuttal PDF file). We believe that these experiments and comparisons serve to demonstrate the effectiveness of our method.
>
>
>
> [1] Sagar Vaze, et al. Generalized category discovery. CVPR, 2022.
>
> [2] Ziwei Liu, et al. Large-scale long-tailed recognition in an open world. CVPR, 2019.
>
> [3] G. Van Horn, et al. The inaturalist species classification and detection dataset. CVPR, 2018.
>
> [4] Kiat Chuan Tan, et al. The herbarium challenge 2019 dataset. In Workshop on Fine-Grained Visual Categorization, 2019.

---

> ### Author Response · Authors · 2023-08-19
> **Comment by Authors**
>
> We deeply appreciate your time and the thoughtful insights you've shared! Based on your comments, we have undertaken a comprehensive revision of our work (please refer to the rebuttal section for details). We hope we have addressed the concerns you raised regarding the originality of the task and approach, as well as the experiments across various datasets.
>
> As the discussion phase is nearing its conclusion, we would be grateful if you would inform us of any remaining concerns or questions you might have. We are more than willing to provide further assistance in clarifying any issues.
>
> Once again, we extend our heartfelt gratitude for your time and the invaluable suggestions you've provided!

---

### Official Review · Reviewer_kmyS · 2023-07-11

**Soundness:** 2 fair
**Presentation:** 2 fair
**Contribution:** 2 fair
**Rating:** 5
**Confidence:** 4

**Summary:**

This paper studies the long-tailed recognition in the presence of open-set samples, which is never studied by previous works. Moreover, existing long-tailed learning methods can not be directly extended to open-set classification. To solve this problem, the authors design a new method termed BaCon, which utilizes contrastive learning to help estimate the label distribution and generate pseudo-labels for the unlabeled data, thus relieving the data imbalance and providing more data supervision. Besides, a pseudo-label-based contrastive loss is proposed to cluster similar samples for better open-set classification. The authors conduct experiments on multiple long-tailed datasets including CIFAR10/100-LT, ImageNet100-LT, and Places-LT. The results show that BaCon can obviously improve performance, especially when the imbalance ratio is high.

**Strengths:**

1. This paper firstly studies the long-tailed recognition with unlabeled data in both open-set and close-set samples. This studied problem is more complex and the performances of most previous methods are not satisfactory, indicating the necessity of proposing a corresponding method.

2. The authors propose a new method for long-tailed open-set classification. The experimental results demonstrate the superiority of the proposed method compared to most existing methods.

3. The authors conduct several ablation studies to verify the effectiveness of each component.

**Weaknesses:**

1. The studied problem is novel. However, the motivation is insufficient. By simply integrating open-set classification methods with re-balancing strategies, the long-tailed problem may be alleviated. The authors should also compare with the simply integrated baseline methods, or explain the difficulty of the integration.

2. The pseudo-labeling module is important since it can affect the additional supervision and the pseudo-label-based contrastive loss. So how to ensure the quality of the pseudo-label? What if the pseudo-label is incorrectly assigned? To verify the quality of the pseudo-label, I suggest the authors calculate the accuracy of the pseudo-labels for both head and tail classes and report the results.

3. Some typos. The title of Section 5 should be "Experiments". In Table 4, GCD performs better in Many and Med shots, but the results are not bolded.

**Questions:**

See "Weaknesses".

**Limitations:**

The authors have discussed the limitations in the main paper.

---

> ### Author Rebuttal · Authors · 2023-08-10
>
> > The studied problem is novel. However, the motivation is insufficient. By simply integrating open-set classification methods with re-balancing strategies, the long-tailed problem may be alleviated. The authors should also compare with the simply integrated baseline methods, or explain the difficulty of the integration.
>
> Thanks for the advice! We would like to claim that most supervised or semi-supervised long-tail learning methods [1, 2, 3] require the prior training set (long-tailed) distribution to tackle the data imbalance issue, and it prevents them from being directly extended to resolve the proposed LT-OPC task, since the prior dataset distribution is unavailable in LT-OPC (line 56 - line 58).
>
> Meanwhile, though several self-supervised methods [4, 5] could alleviate the imbalance issue without knowing the dataset distribution, they discard all the accessible label information and bring marginal accuracy gain. As shown in Table C (please kindly refer to the rebuttal PDF file), we combine two best-performing baseline methods GCD and SimGCD with  BCL [5], which is one of the latest self-supervised long-tail learning methods. It's observed that BCL can not effectively tackle the data imbalance issue, while the proposed BaCon improves both accuracy and balancedness by a large margin via dynamic distribution estimation (Section 4.1) and the self-balanced knowledge transfer module (Section 4.2).
>
> > The pseudo-labeling module is important since it can affect the additional supervision and the pseudo-label-based contrastive loss. So how to ensure the quality of the pseudo-label? What if the pseudo-label is incorrectly assigned? To verify the quality of the pseudo-label, I suggest the authors calculate the accuracy of the pseudo-labels for both head and tail classes and report the results.
>
> Thanks for the advice!  We report the test accuracy of the pseudo-labeling branch during training in Table D (please kindly refer to the rebuttal PDF file). Furthermore, we have two elaborately designed components in our method to ensure the quality of the pseudo-label-based contrastive loss. Firstly, to mitigate the impact of the long-tail distribution on pseudo-labels, we designed a debiasing module to post-hoc adjust the logits output from the pseudo-labeling branch (introduced in Section 4.2). Secondly, when calculating the *positiveness* coefficient $w_{ij}$ using the rectified pseudo-labels, we utilized the soft labels of each sample instead of hard labels (i.e., converting labels to one-hot labels based on the class with the highest logits). This design helps alleviate the influence of erroneous pseudo-labels. Empirical evidence supporting the effectiveness of these two components can be found in Table 6 (b) and (c).
>
> > Some typos. The title of Section 5 should be "Experiments". In Table 4, GCD performs better in Many and Med shots, but the results are not bolded.
>
> We would like to thank the reviewer for a very detailed review! We will fix the typos.
>
> [1] Menon, Aditya Krishna, et al. "Long-tail learning via logit adjustment." ICLR 2021.
>
> [2] Kim, et al. "Distribution aligning refinery of pseudo-label for imbalanced semi-supervised learning." NIPS 2020.
>
> [3] Lee, et al. "ABC: Auxiliary balanced classifier for class-imbalanced semi-supervised learning." NIPS 2021.
>
> [4] Jiang, Ziyu, et al. "Self-damaging contrastive learning." ICML, 2021.
>
> [5] Zhou, Zhihan, et al. "Contrastive Learning with Boosted Memorization." ICML, 2022.

---

> > ### Comment · Reviewer_kmyS · 2023-08-17
> >
> > Thank you for your responses. After reading your rebuttal and the other reviewers' comments, my concerns have been addressed to some extent. I would like to maintain my score.

---

> > > ### Author Response · Authors · 2023-08-18
> > > **Further Reply to Reviewer kmyS**
> > >
> > > Thank you for your feedback! If you have any other questions or concerns, please feel free to reach out to me at your convenience.
> > >
> > > Once again, I sincerely appreciate your time and valuable suggestions!

---

### Official Review · Reviewer_JDMB · 2023-07-26

**Soundness:** 3 good
**Presentation:** 4 excellent
**Contribution:** 3 good
**Rating:** 7
**Confidence:** 4

**Summary:**

This paper introduces and provides a formal definition for a real-world task, the long-tailed open-world classification (LT-OPC): it entails generating predictions for old and novel classes within a long-tailed open-world context. The proposed method incorporates a contrastive-learning branch and a pseudo-labeling branch to offer mutual guidance. Specifically, the contrastive-learning branch ensures reliable distribution estimates to standardize the pseudo-labeling branch's predictions. This subsequently directs contrastive learning via self-balanced knowledge transfer via a novel soft contrastive loss. The paper shows its effectiveness compared to methodologies from related fields of imbalanced semi-supervised learning and generalized category discovery.

**Strengths:**

- Clarity: This paper is well-structured and clearly presents the motivation and method.

-  Significance: The problem of long-tailed open-world classification is a significant real-world problem that this paper strives to define and address.

- Originality: The paper introduces an intriguing concept, "soft contrastive learning," which makes it possible to benefit from pseudolabeling and soft labels for contrastive learning. The pseudolabeling branch provides the contrastive branch with classification signals, while the contrastive branch standardizes the distribution of pseudolabeling branch.

- Quality: The supplementary material and experiments offer a detailed analysis, enhancing the paper's value.

**Weaknesses:**

- It is not a weakness, but since the OLTR work already exists, it is better to use a more distinct name for the problem the paper addresses. Especially since this problem is more in line with generalized category discovery, a different name for the problem seems less confusing.
- Ablations studies could be more well explored, and the explanation of the trends seen in Table 6 can be improved and explain these trends with more depth.
- Calling theorem 1 a theorem does not seem mathematically correct. It is solving a constrained optimization problem specific to the method.
- Also, in the proof of theorem 1 in supplemental, is it correct to omit $C$ for a more rigorous consideration because it depends on $w_{ij}$, which depends on $z_i$ and $z_j$s?
Results for other methods are by applying code of other methods to the new datasets. Since methods like ORCA on cell dataset, GCD and SimGCD on Herbarium have a decent performance. Since Herbarium or cell datasets are long-tailed, comparing these methods on the same datasets and report numbers from their paper seems a more fair comparison.
- Although the number of experiments is enough, the numbers have not been explained and reasoned about. Since numbers for different methods have drastic changes, a section to analyze these numbers (at least in supplemental) seems necessary.
- Minor: there are a few mistakes in the tables: (Table 4/known/Many/GCD: 75.9) (Table4/known/Med/SimGCD: 72.8) (Table5/CIFAR10-LT $\rho =150$/ OpenCon: 83.2) (Table5/CIFAR100-LT$\rho =150$/ BaCon-O: 66.8) should be bold

Also, indicating the second-best method or improvement can be beneficial for the reader to see the efficacy of the paper’s contributions.

**Questions:**

1- In Table 4, the Std for ABC, DARP, and other methods are strangely high. Is there any reason for this? Also, why ORCA performs poorly in comparison to other methods? Since the results in their paper are reasonable, there might be something unfair to their method, especially in Table 4. Having 0.3 for "few" classes looks pretty odd compared to the highest number, 65.3. The same goes for SimGCD, which should have close results to GCD because of their high similarity.

2- There is a large gap between the paper's results and the next one in cifar100-LT, especially for new classes, and also, the model has a higher accuracy for new categories than the old ones, which seems quite strange. Is there any particular intuitive reason for this?

3-  The reason behind the paper outperforming other methods with this high margin has yet to be explored and explained well. It is necessary to explain why methods behave the way the paper has reported for each dataset. Also, numbers for each dataset have drastic variances between different methods. It is worthwhile to investigate and explain the reasons for these differences so that they become more comprehensible.

4- While it is interesting to consider common unseen categories, as in Figure 1, however, the real-world data usually has the long-tailed categories as unseen. With this new way of considering the open world, a model that assigns most novel samples to the most common novel category or, in other words, detects the open world samples and then assigns them to the most common category will triumph over a model which considers the unknown categories all in long-tailed. For instance, ORCA has this uniform objective loss, which is in direct contrast with the way data has been assigned (maybe this is the reason for ORCA's poor performance in the tables), while for the real world, the more traditional long-tailed split is in fact more plausible. Although Data preparation is explained in the paper, is it similar to how Figure 1 has depicted? Are there any experiments to also compare methods against traditional long-tailed versions of datasets?
Also, a base method that only detects open-world samples and assigns them to the most common novel categories might help to decipher how much of this assignment is just because of the model's knowledge of common novel categories.


**Limitations:**

The limitations of the work have been adequately addressed.

---

> ### Author Rebuttal · Authors · 2023-08-10
>
> > A different name for the problem seems less confusing.
>
> Thanks for the advice! We will change the name to 'Imbalanced Open-World Classification'.
>
> > Ablations studies could be more well explored.
>
> Thanks for the advice! We will add a more detailed analysis in the ablation.
>
> > Calling theorem 1 a theorem does not correct.
>
> Thanks for the advice! We will change 'Theorem 1' to 'Claim 1'.
>
> > Is the proof of theorem 1 correct to omit C?
>
> Thank you for your reminder! We will modify the $C$ in Eq. 2 (Appendix) to $C_i$, as $C$ may vary among different samples. But we believe that this should not affect subsequent proofs, because, for a training batch, its *positiveness* score matrix is obtained by calculating the similarity of the rectified class probability distribution between each sample pair $\boldsymbol{x}_i$ and $\boldsymbol{x}_j$ (line 190- line 192), so each element is a fixed value that does not affect the subsequent differentiation process and can be regarded as a constant.
>
> > Results on Herbarium or Single-Cell.
>
> Thanks for the advice! We compare the proposed BaCon with other methods on Herbarium-19 in Table F in the rebuttal pdf. Our method outperforms baselines on open-set ('New' split) classes and overall accuracy ('All' split) by a large margin.  The authors of ORCA didn't provide the source code for training methods on the Single-Cell dataset, which is a dataset in the field of biology and may need some unique preprocessing or different hyper-parameters. We have emailed the authors to ask for the source code, and we will conduct the experiments on Single-Cell when we get the code from the authors.
>
> > The results have not been fully explained.
>
> Thanks for the advice! We will add the explanation below:
>
> In Table A (in the rebuttal PDF), we compared the proposed BaCon with two SOTA imbalance-SSL methods, ABC and DARP, as well as the six latest open-world recognition methods. Among them, GCD and OpenCon are contrastive-based methods, while ORCA, SimGCD, OpenLDN, and TRSSL are classifier-based methods.  **For the imbalanced-SSL methods,** thanks to their carefully designed semi-supervised learning approaches for long-tail data, they achieve decent performance on known (close-set) classes. However, they perform poorly on novel (open-set) classes because the imbalance-SSL scenario does not consider the presence of open-set samples in the unlabeled data.  **Regarding the open-world recognition methods,** we found that contrastive-based methods outperform classifier-based methods in the LT-OPC scenario. Reasonable explanations can be found in [1], which indicate that self-supervised contrastive training, is more robust to class imbalance than supervised methods. Nevertheless, existing contrastive-based methods struggle to optimize the feature space of unlabeled samples, e.g., GCD only uses a self-supervised contrastive loss for unlabeled data. Moreover, GCD methods lack tailored designs for imbalanced datasets, leading to significant performance degradation when the training set has a long-tailed distribution (as shown in Table 2 in the main text).  **For the proposed BaCon,** we design a dual branch structure that works collaboratively to provide interactive supervision and achieve self-rebalancing. The pseudo-labeling branch is enhanced by the proposed 'dynamic distribution estimation' algorithm for regularizing the predictions, while the contrastive-learning branch takes advantage of the 'self-balanced knowledge transfer' module, which helps to learn a balanced and reasonable feature space. It outperforms baseline methods by tackling the imbalanced recognition task and the open-world challenge at the same time.
>
> > Minors in tables.
>
> We would like to thank the reviewer for a very detailed review. We will fix the mistakes in the experiment tables.
>
> > There might be something unfair in experiments.
>
> Please kindly refer to 'The Fairness of the Experiments' in the 'General Response' column at the top.
>
> > In Table 4, the Std for some methods are strangely high.
>
> As you mentioned, it's observed that classifier-based methods have a relatively large Std than contrastive-based methods (GCD and OpenCon). The phenomenon is aligned with [1], which suggests that self-supervised contrastive training, is more robust to class imbalance than supervised methods. We will add the explanations in the final version.
>
> > Although Data preparation is explained in the paper, is it similar to how Figure 1 has depicted? Are there any experiments to compare methods against traditional long-tailed versions of datasets?
>
> In our paper, we assume that both close-set and open-set categories follow a long-tailed distribution, meaning that both parts of the data contain both common and rare classes. For example, in Figure 1 (a), we assume classes 1-5 are close-set categories and 6-10 are open-set categories, with roughly the same distribution for the two parts. The core idea of setting the problem in this way is that when manually collecting data, we can consider it as randomly sampling some categories from a large (including plenty of classes) real-world long-tailed distribution as the close-set. Similarly, the open-set categories can also be considered as a sampling process in the large long-tailed distribution. For instance, we consider training a species classification model where the close-set data we collect covers species in the forest, while open-set classes can be species on plains or marine organisms. In this case, both closed-set and open-set samples are long-tailed.
>
> In the experiment, we compared BaCon with the semi-supervised long-tail learning methods ABC and DARP. On the other, due to we can't obtain a prior distribution of the training set in LT-OPC, we are unable to compare the proposed method with supervised long-tail methods.
>
>
>
> [1] Hong Liu, et al. Self-supervised learning is more robust to dataset imbalance. ICLR, 2021.

---

> > ### Comment · Reviewer_JDMB · 2023-08-16
> >
> > I thank the authors for providing additional experiments and explanations. Most of my concerns are addressed, and here I mention a few remaining suggestions (or discussions).
> >
> > **Problem Name** 'Imbalanced Open-World Classification' is still ambiguous, and it can be mistaken by detecting open-world samples drawn from a long-tailed distribution. Since the method overlaps with generalized category discovery, it seems more in line with the literature if it makes it clear that it is generalized category discovery for a scenario that both novel and seen categories obey long-tailed. Because by looking at this name, the problem should be distinguishable from similar scenarios. Also, I think since the paper's version of long-tailed is different from the traditional long-tailed, introducing a term that also distinguishes this different category distribution can aid future works in delineating which problem they are addressing, traditional long-tailed or this double long-tailed for novel and seen categories.
> >
> > **Ablations** It will be more convincing if it is mentioned how the ablations will be explored because it is unclear whether it will be enough. (For instance, in comments)
> >
> > **Fairness of the method** What I meant by the fairness of the method is that in the paper, it is assumed that both novel and seen scenarios obey Pareto distribution. In contrast, other methods do not have this assumption. It gives an unfair advantage to the method. Although I believe the current long-tailed works that consider the novel categories to be in the tail are also unrealistic, but comparing these methods for the same long-tailed definition they used can show how much of the method's ability is due to the Pareto distribution prior knowledge. For instance, Herbarium results for GCD are much lower than their paper, so it will be fair if the result of this method is also compared on the same distribution they've used so that the robustness of the model is tested.
> >
> > **Theorem 1** I still think $C_i$ also can not be discarded as easily as mentioned. However, since its name has changed from theorem 1 to claim 1, it can be less rigorous, so I consider it resolved.
> >
> > **Results Analysis** Although the problem that the paper has addressed is novel since its novelty is limited, the experimental part and effect of the method itself can benefit from more explanation. I appreciate that the authors addressed the observed trends in results generally, and due to limited space, having an in-depth analysis in the rebuttal is not feasible. But I strongly recommend comprehensively analyzing significant gaps or stds in each table in the revised paper since the paper has a rich set of experiments; it is worthwhile to provide some insights for the reader to understand the purpose of each experiment, how each specific dataset affects the results and so on. It is also beneficial to mention which branch of the method addresses the weakness of each specific set of previous works (open-world recognition and Imbalanced SSL). But in general, I appreciate the provided explanations in the rebuttal, and it provides some insights about why the other methods fail. It is yet to be mentioned *how* the proposed method resolves previous works' shortcomings. It can be speculated which part of the framework addresses each prior work's weakness. However, it will be reassuring if the authors discuss this in the paper (or supplemental).
> >
> > The following paper also might provide some insights about the proposed problem, so it will be helpful to consider mentioning the differences. Can your method be applicable when novel category distribution is arbitrary? I think the branch of your method that makes it robust to category frequencies might also make it distribution-agnostic, similar to [a] scenario (but for generalized category discovery).
> >
> > [a] Yang, Muli, et al. "Bootstrap Your Own Prior: Towards Distribution-Agnostic Novel Class Discovery." Proceedings of the IEEE/CVF Conference on Computer Vision and Pattern Recognition. 2023.

---

> > > ### Author Response · Authors · 2023-08-16
> > > **Further Reply to Reviewer JDMB (1/3)**
> > >
> > > > Problem Name
> > >
> > > Thanks for the advice! Inspired by [a], we will change the name of the setting to 'Distribution-Agnostic Generalized Category Discovery' to better fit our problem setting and reduce ambiguity.
> > >
> > >
> > >
> > > > It will be more convincing if it is mentioned how the ablations will be explored because it is unclear whether it will be enough.
> > >
> > >
> > >
> > > **The effectiveness of ${\mathcal{L}}_{reg}$** Recall in Section 4.1, we propose to regularize the predictions of the pseudo-labeling branch by the estimated train-set distribution. In Table 6a, we show the performance of the pseudo-labeling branch on ImageNet-100-LT with different estimation strategies. 'Oracle' denotes we use the true distribution $\boldsymbol{\pi}$ of $\mathcal{D}$ (unknown in practice) as the target distribution in Eq. 2, and it serves as an upper bound of the performance.  Compared to previous works (ORCA and SimGCD) that use a balance prior, regularizing the predictions with oracle long-tailed distribution significantly improves the performance on both known and novel categories, showing the importance of the distribution estimation process. Meanwhile, the similar results achieved by our estimation strategy imply $\boldsymbol{\pi_e}$ could be a reliable proxy to $\boldsymbol{\pi}$. Furthermore, we investigate whether two alternative estimation strategies could help the pseudo-labeling branch: 1), only regularize known classes prediction with $\boldsymbol{\pi}_{\mathcal{D}^l}$ 2), perform k-means clustering on the feature of the pseudo-labeling branch, and they all results in inferior accuracy and could in turn deteriorate the contrastive learning process. In conclusion, the results indicate that an accurate estimation of the training set distribution is crucial for classifier-based methods (e.g., ORCA and SimGCD) to have decent performance when generalizing to LT-OPC.
> > >
> > > **The effectiveness of sampling & debiasing** In Section 4.2, we suggest adjusting the prediction logits according to the estimated distribution for debiasing and sampling unlabeled instances for re-balancing. We adopt step-by-step ablation experiments to the proposed approaches in Table 6b. 'Baseline' refers to not using the proposed pseudo-label-based contrastive loss, and it leads to inferior performance on novel classes since they only get supervision from the self-supervised contrastive loss. 'Vanilla' means incorporates the designed pseudo-label-based loss into the optimization objective with Eq. 7 without the debiasing and sampling module, i.e., calculate the designed loss directly without performing any preprocessing on pseudo labels. It brings significant performance gain on novel classes (~10%) due to the leverage of additional supervision in pseudo labels. On the other, known classes also benefit from the 'knowledge distillation' process, which implies the probability distribution information from the pseudo-labeling branch is complementary to the one-hot label. While we observe both 'debiasing' and 'sampling' modules could further bring performance gains from the results, and by combining the two techniques together (as described in Section 4.2), we achieve higher test accuracy on both known and novel classes.
> > >
> > > **Definition of the pseudo-label based contrastive loss** In Section 4.3, we design a novel *soft* contrastive loss based on pseudo-labels to transfer the knowledge of $f_{cls}$ into $f_{con}$. As an opposite, we could also construct the loss in a *hard* manner where we formulate the positive pairs on top of the prediction class with the largest logit and further perform the supervised contrastive loss. Intuitively, the *hard* design discards the probability distribution information and is more susceptible to the potential false pseudo-labels, while the *soft* contrastive loss utilized in our method could help alleviate the influence of erroneous pseudo-labels. The empirical results also support the intuition, as shown in Table 6c, the proposed $\mathcal{L}_{CL}^{s}$ (soft) outperforms the supervised CL loss (hard) by a large margin. This phenomenon is also observed in knowledge distillation [1] that transferring knowledge by using soft labels rather than one-hot predictions achieves better performance.
> > >
> > >
> > >
> > > [1] Hinton, et al. "Distilling the knowledge in a neural network." arXiv (2015).

---

> > > > ### Author Response · Authors · 2023-08-16
> > > > **Further Reply to Reviewer JDMB (2/3)**
> > > >
> > > > > **Fairness of the method** What I meant by the fairness of the method is that in the paper, it is assumed that both novel and seen scenarios obey Pareto distribution. In contrast, other methods do not have this assumption. It gives an unfair advantage to the method. Although I believe the current long-tailed works that consider the novel categories to be in the tail are also unrealistic, but comparing these methods for the same long-tailed definition they used can show how much of the method's ability is due to the Pareto distribution prior knowledge. For instance, Herbarium results for GCD are much lower than their paper, so it will be fair if the result of this method is also compared on the same distribution they've used so that the robustness of the model is tested.
> > > >
> > > > Thanks for the advice! We do not sample the Herbarium dataset to a subset with Pareto distribution, and the reason that GCD has lower results is we change the evaluation metric to be more reasonable. Specifically, since the test set of Herbarium is also imbalanced, it could lead to model bias on frequent classes by calculating the accuracies by $Acc = \frac{\textit{number of correct predicted samples}}{\textit{number of all samples}}$ as in GCD, because a model only focuses on frequent categories while totally ignore tail classes is able to obtain high overall accuracy. Hence, we first calculate each class's accuracy and then average across all categories as the final accuracy, which is aligned with the evaluation metric in long-tail learning (in long-tail learning, the test set is balanced, therefore the overall accuracy is equal to the mean accuracy of all categories). In this way, we give equal treatment to all categories and encourage learning a more balanced and unbiased model.
> > > >
> > > > On the other, we have considered modifying existing methods in GCD to generalize them to the proposed LT-OPC scenario. However, our experimental results indicate that improving these methods via long-tail learning approaches cannot achieve satisfying performance. Specifically, we replace the uniform distribution prior (on both closed and open categories) used in SimGCD to the **oracle distribution** of the long-tailed dataset in LT-OPC, which means the distribution of both labeled and unlabeled data, and it's not available in practice. Then, we evaluate the performance of SimGCD equipment with the oracle distribution, which can be regarded as the upper-bound performance for SimGCD, and the results are summarized below:
> > > >
> > > > | Method        | Many$\uparrow$ | Med$\uparrow$ | Few$\uparrow$ | Std$\downarrow$ | All$\uparrow$ | Many$\uparrow$ | Med$\uparrow$ | Few$\uparrow$ | Std$\downarrow$ | All$\uparrow$ | Overall  |
> > > > | ------------- | :------------: | ------------- | :-----------: | :-------------: | :-----------: | :------------: | :-----------: | :-----------: | :-------------: | :-----------: | :------: |
> > > > | SimGCD        |      72.8      | 71.1          |     34.6      |      17.6       |     59.8      |      32.9      |     25.8      |     15.5      |       7.1       |     24.8      |   52.8   |
> > > > | SimGCD-Oracle |      69.4      | **72.9**      |     45.6      |      12.1       |     62.9      |      47.0      |     43.9      |     36.8      |       4.3       |     42.7      |   58.9   |
> > > > | BaCon-O       |      72.9      | 64.7          |   **62.0**    |     **4.6**     |     66.5      |      72.6      |     70.6      |   **65.3**    |     **3.1**     |     69.6      |   67.1   |
> > > > | BaCon-S       |    **73.5**    | 65.6          |     61.2      |       5.1       |   **66.8**    |    **74.7**    |   **73.1**    |     64.5      |       4.5       |   **71.0**    | **67.6** |
> > > >
> > > > It's observed that the performance of SimGCD still has an inferior performance on LT-OPC, even though we have provided the oracle distribution to it for regularization, while the proposed BaCon outperforms them by a large margin. It indicates that simply modifying methods in GCD with re-balancing techniques can not perform well in LT-OPC. While BaCon outperforms baselines by the designing of the dual branch structure which works collaboratively to provide interactive supervision and achieve self-rebalancing, also, the contrastive-learning branch takes advantage of the 'self-balanced knowledge transfer' module, which helps to learn a balanced and reasonable feature space.

---

> > > > > ### Author Response · Authors · 2023-08-16
> > > > > **Further Reply to Reviewer JDMB (3/3)**
> > > > >
> > > > > > The following paper also might provide some insights about the proposed problem, so it will be helpful to consider mentioning the differences. Can your method be applicable when novel category distribution is arbitrary? I think the branch of your method that makes it robust to category frequencies might also make it distribution-agnostic, similar to [a] scenario (but for generalized category discovery).
> > > > >
> > > > > Thanks for bringing up the enlightening work. Our method could be applicable in distribution-agnostic scenarios. Similar to the mentioned paper [a], we also evaluate the proposed method on datasets with different imbalance ratios (Table 5 in the main text). Furthermore, we verify the effectiveness of BaCon when the imbalance ratio of the labeled and unlabeled subsets are different in Appendix E. In addition, results with different known classes $\vert \mathcal{Y}_k \vert$ and results on the few annotated scenarios are also presented in Appendix E, the proposed method consistently outperforms baselines which indicate our method is robust to the change of dataset distribution or composition.
> > > > >
> > > > >
> > > > >
> > > > > [a] Yang, Muli, et al. "Bootstrap Your Own Prior: Towards Distribution-Agnostic Novel Class Discovery." CVPR, 2023.

---

> > > > > ### Comment · Reviewer_JDMB · 2023-08-16
> > > > >
> > > > > These new explanations are much more illuminating. Including these in the paper can eradicate many questions, and I appreciate the authors' new explanations and their effort in providing details. I suggest making it more evident in the paper about the strengths of your method in being robust to different distributions because calling your method long-tailed might not delineate the novelty that the problem suggests from the already established OLTR. I appreciate these new clarifications, and I believe the paper will be more complete by including the said changes; Hence, I update my score from 6 to 7.

---

> > > > > > ### Author Response · Authors · 2023-08-17
> > > > > > **Further Reply to Reviewer JDMB**
> > > > > >
> > > > > > Thank you for these valuable suggestions and valuable feedback, which have helped us further improve and enhance the quality of our work.  We will thoroughly revise our paper based on your suggestion.
> > > > > >
> > > > > > We've noticed that the score is still at 6. Would you mind updating it at your convenience? Thanks again for the appreciation and your valuable time!

---

> > > > ### Comment · Reviewer_JDMB · 2023-08-16
> > > >
> > > > I thank the authors for their additional refinements to the paper. I find this new name much more precise and explains better what is the problem that the paper is addressing. Also, these new ablations shed more light on the different aspects of the proposed method.

---

### Author Rebuttal · Authors · 2023-08-10

# General Response

## To All Reviewers.

Dear Reviewers:

We would like to thank you for your time and insightful comments! We have carefully read your review comments and conducted additional experiments as required to answer the questions (please kindly refer to the rebuttal PDF file and the rebuttal reply to each reviewer below). We hope we have addressed your concerns. We would be grateful if you would kindly let us know of any other concerns and if we could further assist in clarifying any other issues.

Thanks a lot again, and with sincerest best wishes

Authors

## Explanation of the Rebuttal PDF.

In the rebuttal PDF file, we indicate the best performance among **all** methods with **bold numbers** in all tables (consistent with the submitted version), and we will use underlined numbers to represent the best performance among the **baseline** methods. We believe this will help highlight the differences between our method and the baseline methods.

## The Novelty of the Proposed Setting.

GCD [1] and OLTR [2], are closely related to the proposed LT-OPC setting, but we would like to clarify that the proposed problem setting still has significant differences from them and introduces unique challenges in this scenario. **Compared to GCD,** they only focus on the problem of the open world but ignore the data imbalance issue in the real world, i.e., their method falls in the lower right corner of Figure 1 (b); while our setting aims to solve both the problems of the open-world and data imbalance, corresponding to the upper right corner of Figure 1 (b). Moreover, since we do not know the prior distribution of the dataset in open-world scenarios, we cannot simply transfer existing long-tail learning methods to solve the problem of data imbalance in LT-OPC scenarios, making it more challenging. **Compared to OLTR,** though we both study the long-tail problem in open-world scenarios, our method pushes 'one step further'. That is, not only learn a model to detect the open-set samples, but it also requires the model to classify them according to their semantics. On the other, OLTR considers a fully-supervised scenario that only utilizes manually annotated samples (lower left in Figure 1 (c)); while we also make use of unlabeled (close-set and open-set) data (upper right and lower right in Figure 1 (c)), which makes our approach more powerful and generalizable.

The experimental results empirically validate the highly challenging nature of the proposed setting. GCD methods as well as long-tailed SSL methods, all exhibit significant performance drops under the newly proposed setting, as shown in Table 2 in the main text, and Table A, Table B in the rebuttal PDF file.

Additionally, another significance of our work is that existing research on long-tailed learning and open-set learning has developed independently, and very few previous works attempt to combine both of these areas. However, data imbalance and open-ended distribution are inherently intertwined with each other in the real visual world, which renders existing methods ineffective in terms of deployment. The proposed LT-OPC addresses both of these problems simultaneously, which can facilitate the deployment of existing methods in practical application scenarios.

## The Fairness of the Experiments.

In the submitted version, we have kept the original settings of the baseline methods. Namely, ABC, DARP, Opencon, and ORCA, which use the ResNet backbone model as they implement in their papers. On the other hand, GCD, SimGCD, and our method utilize the DINO pre-trained ViT network, which may lead to an unfair comparison. To address this issue, **we have replaced the backbone networks of all baseline methods with the DINO pre-trained ViT, consistent with the proposed method**. Furthermore, we have ensured that all methods are trained for 200 epochs, with an equal amount of data used per epoch. Table A and Table B in the rebuttal PDF file present the modified versions of Table 3 and Table 4 in the main text, respectively, where $\dagger$ denotes adapted methods. It can be observed that the baseline methods benefit from the powerful feature extraction capability of the DINO pre-trained ViT, achieving higher accuracy than using ResNet as the backbone on all datasets. At the same time, our method still demonstrates significant performance improvement.

In the final version, we will replace the original tables and report the experimental results of all methods when using the DINO pre-trained ViT as the backbone.



[1] Sagar Vaze, et al. Generalized Category Discovery. CVPR, 2022.

[2] Ziwei Liu, et al. Large-scale long-tailed recognition in an open world. CVPR, 2019.

---

### Author Response · Authors · 2023-08-15
**General Response to All Reviewers**

Dear Reviewers:

Thank you again for your time and insightful comments! We have comprehensively revised our work according to your comments (please kindly refer to the rebuttal below). We hope we have addressed your concerns regarding the fairness of experiments, novelty of the task and method, etc. **Since the discussion is about to close, we would be grateful if you would kindly let us know of any other concerns and if we could further assist in clarifying any other issues.**

Thanks a lot again, and with sincerest best wishes

Authors

---

### Comment · Area_Chair_7YCm · 2023-08-21
**Final Rating Required**

Dear Reviewer udGd,

Could you help share a quick feedback to authors' rebuttal and give your final rating? Thank you!

Best regards,
AC

---

### Decision · Program_Chairs · 2023-09-21

**Decision:**

Accept (poster)

**Comment:**

This was a slightly positive “borderline accept” paper. Initially, there were common concerns on the novelty of the proposed setting and the proposed method. The submitted paper was not presented well in highlighting the difference, key challenge, and key component of the proposed setting and method. However, after a deep multi-round discussions between authors and Reviewers JDMB (from 6 -> 7) and 7Jpa (from 3 -> 6), the contributions of the paper have been made much clearer. Yet it is unfortunate that Reviewer udGd (4: Borderline reject; Confidence: 3), who also had critical concerns about the novelty of problem setting and proposed method, did not come back after authors’ rebuttal. However, from authors’ feedback and the above-mentioned discussions, AC regarded that these critical concerns have been addressed. There were a number of other concerns, such as experimental comparisons, which were also addressed by authors. Finally, except Reviewer udGd who held the only negative rating but did not come back, all other review ratings were positive, recommending acceptance of the paper. AC therefore is happy to accept the paper. Authors are required to address reviewer comments and incorporate the rebuttal and discussion material to the camera-ready version of the paper.